# Zinc finger protein Zfp335 controls early T-cell development and survival through β-selection-dependent and -independent mechanisms

Xin Wang[1,2†], Anjun Jiao[1,2†], Lina Sun[1,2†], Wenhua Li[1,2†], Biao Yang[1,2], Yanhong Su[1,2], Renyi Ding[1,2], Cangang Zhang[1,2], Haiyan Liu[1,2], Xiaofeng Yang[1,2,3,4], Chenming Sun[1,2,4], Baojun Zhang[1,2,3,4]*

[1]Department of Pathogenic Microbiology and Immunology, School of Basic Medical Sciences, Xi'an Jiaotong University, Xi'an, China; [2]Institute of Infection and Immunity, Translational Medicine Institute, Xi'an Jiaotong University Health Science Center, Xi'an, China; [3]Key Laboratory of Environment and Genes Related to Diseases (Xi'an Jiaotong University), Ministry of Education, Shaanxi, China; [4]Xi'an Key Laboratory of Immune Related Diseases, Shaanxi, China

*For correspondence:
bj.zhang@mail.xjtu.edu.cn

†These authors contributed equally to this work

Competing interest: The authors declare that no competing interests exist.

**Abstract** T-cell development in the thymus undergoes the process of differentiation, selective proliferation, and survival from CD4⁻CD8⁻ double negative (DN) stage to CD4⁺CD8⁺ double positive (DP) stage prior to the formation of CD4⁺ helper and CD8⁺ cytolytic T cells ready for circulation. Each developmental stage is tightly regulated by sequentially operating molecular networks, of which only limited numbers of transcription regulators have been deciphered. Here, we identified Zfp335 transcription factor as a new player in the regulatory network controlling thymocyte development in mice. We demonstrate that *Zfp335* intrinsically controls DN to DP transition, as T-cell-specific deficiency in *Zfp335* leads to a substantial accumulation of DN3 along with reduction of DP, CD4⁺, and CD8⁺ thymocytes. This developmental blockade at DN stage results from the impaired intracellular TCRβ (iTCRβ) expression as well as increased susceptibility to apoptosis in thymocytes. Transcriptomic and ChIP-seq analyses revealed a direct regulation of transcription factors *Bcl6* and *Rorc* by Zfp335. Importantly, enhanced expression of TCRβ and *Bcl6/Rorc* restores the developmental defect during DN3 to DN4 transition and improves thymocytes survival, respectively. These findings identify a critical role of *Zfp335* in controlling T-cell development by maintaining iTCRβ expression-mediated β-selection and independently activating cell survival signaling.

## Editor's evaluation

The authors have discovered that the transcription factor Zfp335 is an important regulator of early T cell development in the thymus. This paper will be of interest to scientists within the field of T cell development. The authors show that Bcl6 and Rorc are direct gene targets of Zfp335 and dysregulation of these are at least partly responsible for the impaired T cell development in Zfp335 mice.

## Introduction

T-cell development proceeds in a series of developmental stages, which is precisely orchestrated by multiple signaling and molecular networks (*Hosokawa and Rothenberg, 2021*; *López-Rodríguez et al., 2015*; *Rothenberg, 2014*). Prethymic progenitor cells originated from bone marrow migrate

into the thymus and sequentially differentiate into CD4⁻CD8⁻ (DN), CD4⁺CD8⁺ (DP), and the CD4⁺ or CD8⁺ (SP) stage. Based on the expression of CD44 and CD25, DN thymocytes are divided into several phenotypically distinct stages, including DN1 to DN4 (*Rothenberg et al., 2008*; *Yang et al., 2010*; *Yui and Rothenberg, 2014*; *Kurd and Robey, 2016*). In the presence of Notch signaling, early thymic progenitor (ETP)-DN1 cells transit into DN2a stage, initiating the T-cell lineage commitment, which is immediately accompanied by TCRβ gene arrangement. The majority of DN2 cells enter the DN3 stage with αβ lineage potential (*Godfrey et al., 1993*). Only DN3 cells with a complete pre-TCR complex, which consists of the functional TCRβ protein, pre-Tα (pTα) chain, and CD3 molecule, can successfully trigger the subsequent maturation into DN4 and DP thymocytes. Further differentiation into mature CD4⁺ or CD8⁺ T cells requires positive and negative selection at DP stage before they migrate to peripheral lymphoid organs (*Dudley et al., 1994*; *Hoffman et al., 1996*; *von Boehmer and Fehling, 1997*; *Malissen et al., 1999*).

Pre-TCR signals regulate thymocyte differentiation by mediating protection from apoptosis, stimulating proliferation, and inducing allelic exclusion at the TCRβ locus in post-β-selection DN3b cells and promoting DN to DP transition (*Hoffman et al., 1996*; *Aifantis et al., 1997*; *Kruisbeek et al., 2000*). Inactivation of pre-TCR components dampens T-lymphocyte development by arresting thymocytes at the DN3 stage and inducing apoptosis (*Fehling et al., 1995*; *Malissen et al., 1995*; *Mombaerts et al., 1992a*; *Mombaerts et al., 1992b*). Multiple transcription factors downstream of pre-TCR signaling are involved in T-cell differentiation and survival. The major pre-TCR signaling is conducted through the dose-dependent expression of Notch controlled by the Id3–E2A axis (*Liu et al., 2021*). Abrogation of either Notch or E2A expression may lead to the developmental block of thymocytes at multiple stages (*Ikawa et al., 2006*; *Shah and Zúñiga-Pflücker, 2014*; *Belle and Zhuang, 2014*). In addition, activation of NF-κB (*Voll et al., 2000*), Ets1 (*Eyquem et al., 2004*), and NFAT5 (*Berga-Bolaños et al., 2013*) by pre-TCR signals also contributes to the developmental block. The transcription factor T-cell factor 1 (TCF1), together with its downstream Bcl-11b, not only increases the potential to differentiate into T cells (*Li et al., 2010*; *Ikawa et al., 2010*), but also positively regulates thymocyte development via promoting TCRβ recombination and expression, as well as DP cell survival (*Albu et al., 2007*; *Li et al., 2013*). Apart from the essential role in the T follicular helper cell lineage commitment (*Yu et al., 2009*), Bcl6 induced by pre-TCR signals is also involved in the DN to DP transition and protection of DN4 cells from apoptosis (*Solanki et al., 2020*). Additionally, abrogation of Rorc expression in thymocytes leads to a decreased DP proportion and impaired DP survival in a Bcl-xl-dependent manner (*Villey et al., 1999*; *Sun et al., 2000*; *Xi et al., 2006*).

Although pre-TCR signaling is crucial for the β-selection checkpoint, it is not sufficient for progression to the DN3 stage. Other pathways or transcription factors coupled with or independent of conventional pre-TCR signaling are found to play important roles in the process (*Ciofani et al., 2004*). The developmental blockade in *Smarca5*- or *Nkap*-deficient thymocytes is confirmed by intact pre-TCR signals in these mice (*Pajerowski et al., 2009*; *Zikmund et al., 2019*). Overall, it remains largely unknown which factors are crucial for T-cell development through mechanisms independent of pre-TCR signaling.

Zfp335, also known as the nuclear hormone receptor coregulator (NRC) – interacting factor 1 (NIF-1), is a zinc finger protein with a 13 C2H2 zinc finger repeating structure consisting of 1337 amino acids (*Han et al., 2016*). The C2H2-ZF family encodes more than 700 proteins in the human genome, some of which play important roles in ontogenesis, immune cell differentiation, and disease occurrence (*Li et al., 2010*; *Heizmann et al., 2018*), yet the biological characteristics and functions of most members are unclear (*Stubbs et al., 2011*; *Emerson and Thomas, 2009*). Zfp335 regulates gene transcription by recruiting H3K4 methyltransferase complexes, interacting with coactivators, or directly binding to certain gene promoters (*Han et al., 2016*; *Yang et al., 2012*; *Mahajan et al., 2002*; *Wolfe et al., 2000*). Zfp335 plays important regulatory roles in early embryonic development and neurogenesis (*Yang et al., 2012*). Germline knockout of Zfp335 is embryonic lethal, while deletion of Zfp335 gene in nerve cells impairs the proliferation and differentiation of nerve progenitor cells in mice, eventually leading to severe microcephaly (*Yang et al., 2012*). Function of Zfp335 in T-cell development has been observed in the study of the *Zfp335bloto* allele, a missense mutation derived from ENU mutagenesis. While thymocyte development is not significantly affected by this hypomorph mutation, there is a significant reduction in the number of peripheral T cells due to defects in the

maturation and migration of thymocytes (*Han et al., 2014*). Without loss of function studies, it remains to be determined whether Zfp335 is required for intrathymic T-cell development.

In this study, we investigated Zfp335 expression during different thymocyte stages, as well as its function at the β-selection checkpoint and during DN to DP transition. We found that in the thymus, Zfp335 has the highest expression in DN3 thymocytes. Zfp335 is indispensable for thymocyte β-selection and supports the transition from DN to DP stage by maintaining intracellular TCRβ (iTCRβ) expression, as well as by promoting DN and DP thymocyte survival via directly regulating *Bcl6* and *Rorc* expression.

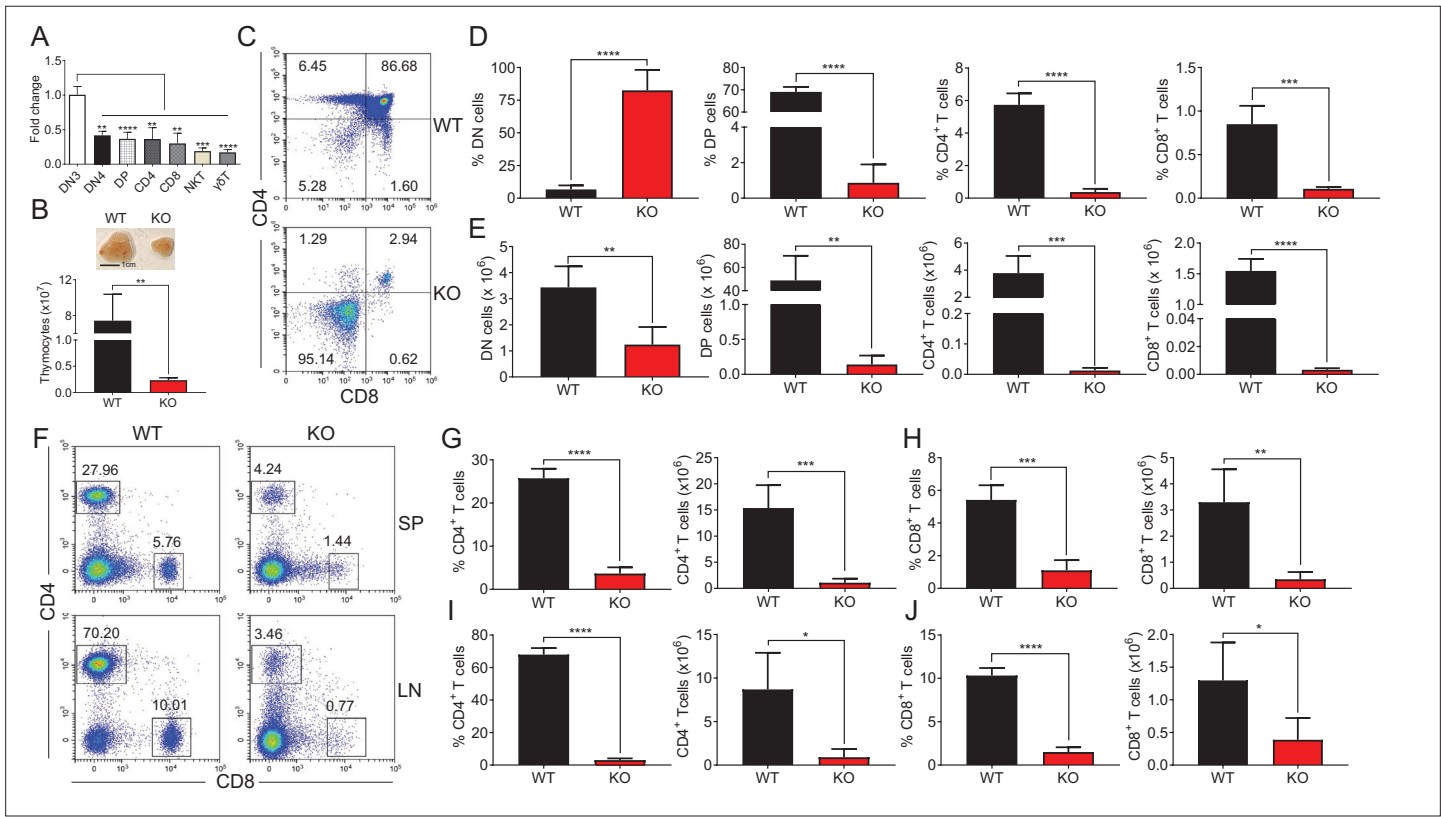

**Figure 1.** Impaired thymocyte development in *Zfp335*-deficient mice. (**A**) DN3, DN4, DP, CD4, CD8, NKT, and γδT cells were sorted from C57BL/6 thymocytes by flow cytometry. The mRNA levels of Zfp335 were measured by qPCR. (**B**) Thymi from *LckCre⁺Zfp335⁺/⁺* (WT) and *LckCre⁺Zfp335ᶠˡ/ᶠˡ* (KO) mice. Representative thymi and total cell number of thymocytes. Scale bar, 1 cm. (**C–E**) The different stages of thymocyte development in WT and KO mice were measured by flow cytometry. (**C**) Representative flow cytometry (FACS) plots of DN, DP, CD4, and CD8 thymocytes. (**D**) The percentages of DN, DP, CD4, and CD8 thymocytes. (**E**) The numbers of DN, DP, CD4, and CD8 thymocytes. (**F–J**) CD4⁺ and CD8⁺ cells in spleen and lymph nodes from WT and KO mice were measured by flow cytometry. (**F**) Representative FACS plots of CD4⁺ and CD8⁺ cells in spleen and lymph nodes. The percentage and number of CD4⁺ T cells (**G**) and CD8⁺ T cells (**H**) in the spleen from WT and KO mice. The percentage and number of CD4⁺ T cells (**I**) and CD8⁺ T cells (**J**) in the lymph nodes from WT and KO mice. Results represent three independent experiments. *n* = 4 mice per group. *p < 0.05, **p < 0.01, ***p < 0.001, and ****p < 0.0001.

The online version of this article includes the following source data and figure supplement(s) for figure 1:

**Source data 1.** *Figure 1D* The percentages of DN, DP, CD4, and CD8 thymocytes from WT and KO mice.

**Figure supplement 1.** Zfp335 protein expression in thymocytes.

**Figure supplement 1—source data 1.** *Figure 1—figure supplement 1B*.

**Figure supplement 2.** The transcriptional profiling of Zfp335 expression in various subsets of thymocytes.

**Figure supplement 3.** Verification of *Zfp335* conditional knockout mouse strain.

**Figure supplement 3—source data 1.** The gel of Zfp335 DNA in WT and KO mice.

## Results

### Impaired thymic αβ T-cell development in *Zfp335*-deficient mice

To study the role of Zfp335 in T-cell development, we first assessed the expression of Zfp335 among different thymocyte subsets, including DN3, DN4, DP, CD4, CD8, NKT, and γδ T cells. We found that DN3 cells displayed a relatively high level of *Zfp335* mRNA expression (*Figure 1A*). Flow cytometry analysis also revealed that Zfp335 protein had the highest expression in DN3 thymocytes (*Figure 1—figure supplement 1A, B*). Consistently, microarray data from ImmGen showed higher expression of *Zfp335* specifically at the DN3a stage during T-cell development from ETP to CD4/CD8 SP (*Figure 1—figure supplement 2A*). Although RNA-seq data from ImmGen exhibited the highest expression of *Zfp335* in DP thymocytes, a gradually increased expression was observed from ETP to DN3 (*Figure 1—figure supplement 2B*). Given the importance of DN3 stage during β-selection checkpoint, we obtained T-cell-specific *Zfp335* mice by crossing *Zfp335^fl/fl* strain with Lck-Cre strain (*Figure 1—figure supplement 3A*). *Zfp335* deletion was confirmed by real-time PCR (qPCR) analysis in DN4 cells (*Figure 1—figure supplement 3B*). Strikingly, *LckCre^+Zfp335^fl/fl* (KO) mice exhibited significantly smaller thymi and drastically decreased thymocyte numbers than WT control (*Figure 1B*). Further analysis showed that both percentages and numbers of DP cells, as well as CD4 SP and CD8 SP cells, were considerably reduced in KO mice (*Figure 1C–E*). Conversely, the percentage of DN cells was increased by nearly 10–15-folds, although the total number was decreased (*Figure 1C–E*). In secondary lymphoid organs, we also observed reduced CD4$^+$ and CD8$^+$ cells in the spleen and lymph nodes (*Figure 1F–J*). Thus, Zfp335 is essential for the development of αβT cells in the thymus.

### Zfp335 intrinsically regulates T-cell development in the thymus

To address whether *Zfp335* deletion intrinsically affects T-cell development, Lin$^-$CD25$^+$CD44$^-$ DN3 cells from WT or KO mice were harvested and plated with OP9-DL1 cells in the presence of Flt3L and IL-7, an in vitro model for T-cell development (*Figure 2A*; *Kondo et al., 2017*). On both days 2 and 4, KO group produced fewer DP cells than WT control (*Figure 2B–D*, *Figure 2—figure supplement 1A*). When DN3 cells from WT (CD45.1$^+$) and KO (CD45.2$^+$) mice were mixed and cocultured at a 1:4 ratio (KO cells were rapidly competed out when used at a 1:1 ratio), significantly lower percentage and cell proportion of DP cells were also observed in KO group (*Figure 2E–G*, *Figure 2—figure supplement 1B*). Furthermore, in vivo T-cell development was investigated by adoptively transferring T- and B-cell-depleted bone marrow cells from WT (CD45.1$^+$) and KO (CD45.2$^+$) mice with a 1:4 ratio into WT (CD45.1$^+$CD45.2$^+$) recipients (*Figure 2H*). When thymocytes were analyzed 6 weeks post transfer, both the percentages and cell numbers of DP, CD4, and CD8 cells were significantly reduced while DN percentage was increased in the mice that received KO cells (*Figure 2I, J*, *Figure 2—figure supplement 1C*). Consistently, when WT and KO bone marrow cells were mixed at a 1:4 ratio for cotransfer experiments, fewer DP cell proportion detected in KO compared to WT group (*Figure 2K, L*; *Figure 2—figure supplement 1D*). In addition, DN3 cells from KO mice also generated fewer γδ T cells, despite a higher percentage, in comparison to WT controls after culturing for 4 days (*Figure 2—figure supplement 2A–C*). Together, we demonstrate that Zfp335 intrinsically regulates thymocyte development from DN to DP stage.

### Loss of *Zfp335* blocks the transition from DN3 to DN4 stage

We further examined the impact of *Zfp335* deletion on DN thymocyte development. Staining of CD44 and CD25 on pregated lineage (CD4/CD8/TCRβ/TCRδ/NK1.1/CD19/CD11b/CD11c)-negative thymocytes was performed in WT and KO mice. The results showed that a higher percentage of CD44$^-$CD25$^+$ DN3 cells but a lower percentage of CD44$^-$CD25$^-$ DN4 cells were found in KO group compared to WT group (*Figure 3A, B*), in which the numbers of both DN3 and DN4 thymocytes were decreased (*Figure 3C*). The developmental blockade from DN3 to DN4 in *Zfp335*-deficient cells was verified by in vitro coculture assays on day 2 using OP9-DL1 cultured with WT and KO DN3 cells, respectively (*Figure 3D, E*) or mixed at a 1:4 ratio (*Figure 3F, G*) as described above. The in vivo bone marrow chimera models further confirmed the development block at the DN3 stage (*Figure 3H–K*), indicating that Zfp335 is indispensable for DN3 to DN4 transition during early-stage differentiation.

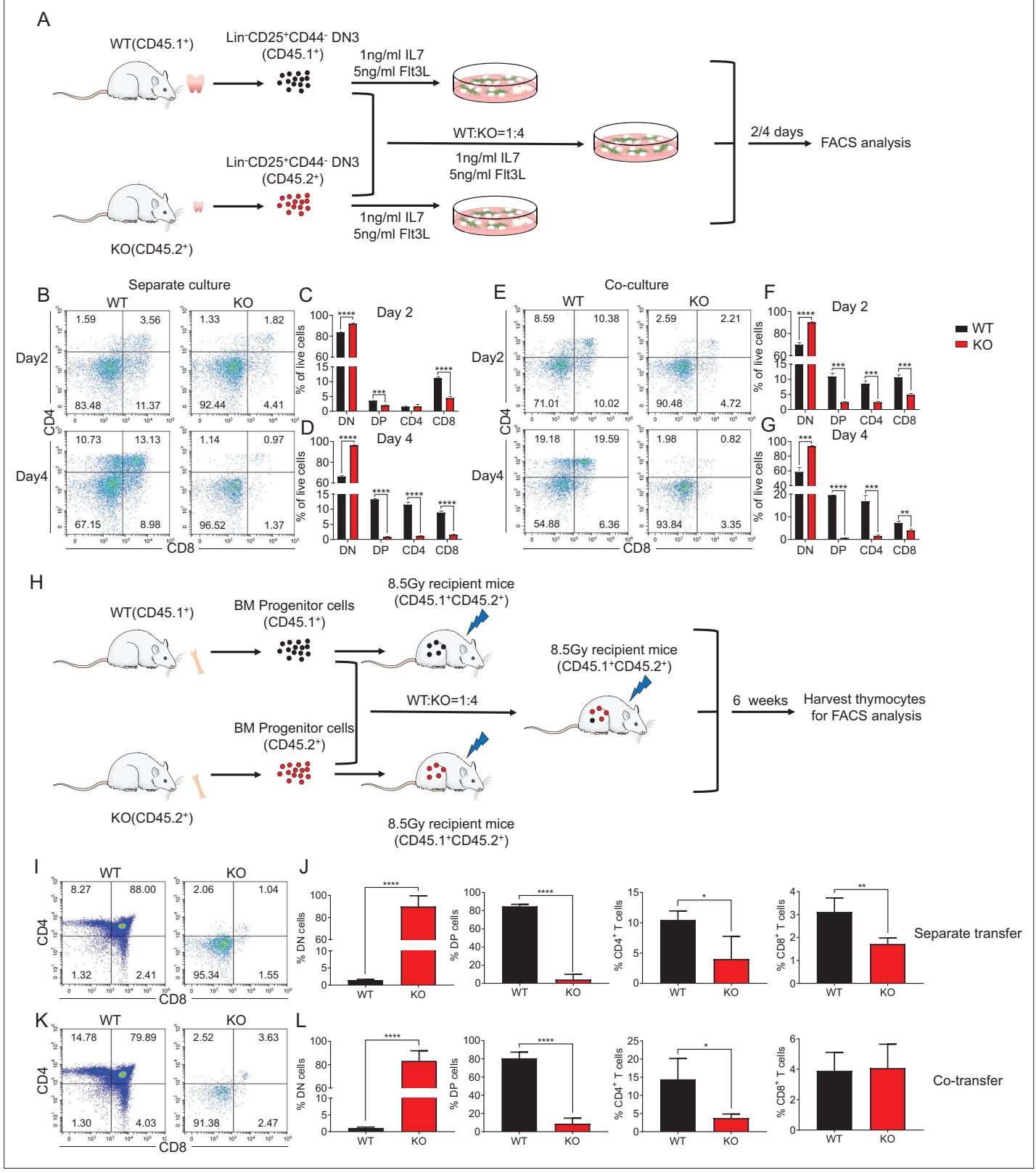

**Figure 2.** An intrinsic block from DN to DP stage in *Zfp335*-deficient mice. (**A**) Schematic overview of the in vitro OP9-DL1 stromal coculture assay for T-cell differentiation from WT (CD45.1[+]) and KO (CD45.2[+]) DN3 thymocytes to DP and SP thymocytes. (**B–D**) WT and KO DN3 thymocytes were cultured with OP9-DL1 feeder cells in vitro in the presence of IL-7 (1 ng/ml) and Flt3L (5 ng/ml) for 2 and 4 days. The DN and DP thymocytes were measured by flow cytometry (*n* = 3). (**B**) Representative FACS plots of DN, DP, CD4[+], and CD8[+] thymocytes. (**C**) The percentages of DN, DP, CD4[+], and

*Figure 2 continued on next page*

*Figure 2 continued*

CD8[+] thymocytes 2 days post culture in vitro. (**D**) The percentage of DN, DP, CD4[+], and CD8[+] thymocytes 4 days post culture in vitro. (**E–G**) A mixed population of WT and KO DN3 thymocytes at a 1:4 ratio was cocultured with OP9-DL1 feeder cells in vitro in the presence of IL-7 (1 ng/ml) and Flt3L (5 ng/ml) for 2 and 4 days. The DN and DP thymocytes were phenotyped by flow cytometry (*n* = 3). (**E**) Representative FACS plots of DN, DP, CD4[+], and CD8[+] thymocytes. (**F**) Percentages of DN, DP, CD4[+], and CD8[+] thymocytes 2 days post culture in vitro. (**G**) Percentages of DN, DP, CD4[+], and CD8[+] thymocytes 4 days postculture in vitro. (**H**) Schematic overview of the in vivo bone marrow chimeric mice model for T-cell differentiation from WT (CD45.1[+]) and KO (CD45.2[+]) progenitors cells to DP and SP thymocytes. (**I, J**) Full chimeric mice were generated by transplanting WT (CD45.1[+]) or KO (CD45.2[+]) bone marrow progenitor cells into lethally irradiated (8.5 Gy) WT recipient mice (CD45.1[+]CD45.2[+]). Six weeks after transplantation, thymi from recipient mice were harvested. (**I**) Representative FACS plots of DN, DP, CD4[+], and CD8[+] thymocytes. (**J**) The percentages of DN, DP, CD4[+], and CD8[+] thymocytes. (**K, L**) Full chimeric mice were generated by transplanting a mixed population of WT (CD45.1[+]) and KO (CD45.2[+]) bone marrow progenitor cells at a 1:4 ratio into lethally irradiated WT recipients (CD45.1[+]CD45.2[+]) with 8.5 Gy. Six weeks after transplantation, thymi from recipient mice were harvested. (**K**) Representative FACS plots of DN, DP, CD4[+], and CD8[+] thymocytes. (**L**) Percentages of DN, DP, CD4[+], and CD8[+] thymocytes. Results represent three independent experiments. *n* = 4 mice per group. *p < 0.05, **p < 0.01, ***p < 0.001, and ****p < 0.0001.

The online version of this article includes the following source data and figure supplement(s) for figure 2:

**Source data 1.** *Figure 2C, D*. The percentages of DN, DP, CD4[+], and CD8[+] thymocytes 2 and 4 days post separate culture in vitro.

**Figure supplement 1.** *Zfp335* deletion caused defects of thymocyte development.

**Figure supplement 1—source data 1.** *Figure 2—figure supplement 1A*. The numbers of DN, DP, CD4[+], and CD8[+] thymocytes 2 and 4 days post separate culture in vitro.

**Figure supplement 2.** Effect of *Zfp335* deletion on γδ T-cell percentages and cellularity in vitro.

**Figure supplement 2—source data 1.** *Figure 2—figure supplement 2B, C*. The percentages and numbers of γδ T cells in WT and KO mice 2 and 4 days post coculture in vitro.

## Ablation of Zfp335 promotes apoptosis in thymocytes

During the β-selection, efficient proliferation of pre-T cells is necessary for DN to DP progression (*Kreslavsky et al., 2012*), during which the pre-TCR signal functions as a positive regulator of thymocyte survival, allowing for differentiation from pre-T cells into DP thymocytes. We sought to examine whether the defect in *Zfp335* KO thymocyte development is due to impaired proliferation or survival. The in vivo BrdU incorporation assay showed comparable or even higher percentages of BrdU[+] DN3 and DN4 cells (*Figure 4A, B*) as well as Ki67[+] DN3 and DN4 cells (*Figure 4—figure supplement 1*). Nevertheless, when we examined thymocyte apoptosis, KO mice showed a remarkably higher percentages of Annexin V[+] DN3 and DN4 cells compared to WT cells (*Figure 4C, D*). After coculture with OP9-DL1 for 4 days, both *Zfp335*-deficient DN3 and DN4 cells showed an increased percentage of Annexin V[+] cells (*Figure 4E, F*). Further examination of the transition between the DN3a and DN3b stages found that the frequencies of both DN3a and DN3b were comparable between WT and *Zfp335*-deficient groups (*Figure 4—figure supplement 2A–C*). However, both cell populations showed enhanced cell apoptosis in *Zfp335*-deficient cells (*Figure 4—figure supplement 2D–F*), indicating that Zfp335 regulates thymocyte apoptosis in a TCR-independent manner. Moreover, in mixed bone marrow chimeras, *Zfp335*-deficient DN3 and DN4 cells also displayed significantly higher Annexin V[+] cells (*Figure 4G, H*), indicating an intrinsic role of Zfp355 in regulating thymocyte survival.

## Ectopic expression of TCRαβ overcomes DN3 stage block in *Zfp335*-deficient mice

The pre-TCR signal-controlled β-selection is essential for thymocyte differentiation from DN3 to DN4 stage (*Zhao et al., 2019*). We then examined the expression of genes involved in the pre-TCR complex in DN3 and DN4 cell populations. The expression of *Ptcra* (encoding pTα) was slightly increased while *Trbc1*, *Trbc2*, and *Cd3e* in DN3 and DN4 cells were comparable in WT and KO group (*Figure 5—figure supplement 1A, B*). Additionally, the expression of intracellular CD3 (iCD3) was also unaffected in both *Zfp335*-deficient DN3 and DN4 cells (*Figure 5—figure supplement 2A, B*). Interestingly, flow cytometry analysis showed no substantial difference in iTCRβ expression between WT and KO DN3 cells, whereas *Zfp335*-deficient mice displayed a significant decrease in the percentage of iTCRβ[+] DN4 cells (*Figure 5A, B*). To further address whether Zfp335 deficiency affects TCRβ expression, we compared differential usage of TCRβ in DN4 cells between WT and KO mice, in which the expression of TCR Vβ5, Vβ6, Vβ8, and Vβ12 were decreased concomitantly (*Figure 5C–F*). However, genomic DNA analysis for V-DJβ5 and V-DJβ8 rearrangements showed no difference between WT and KO DN3

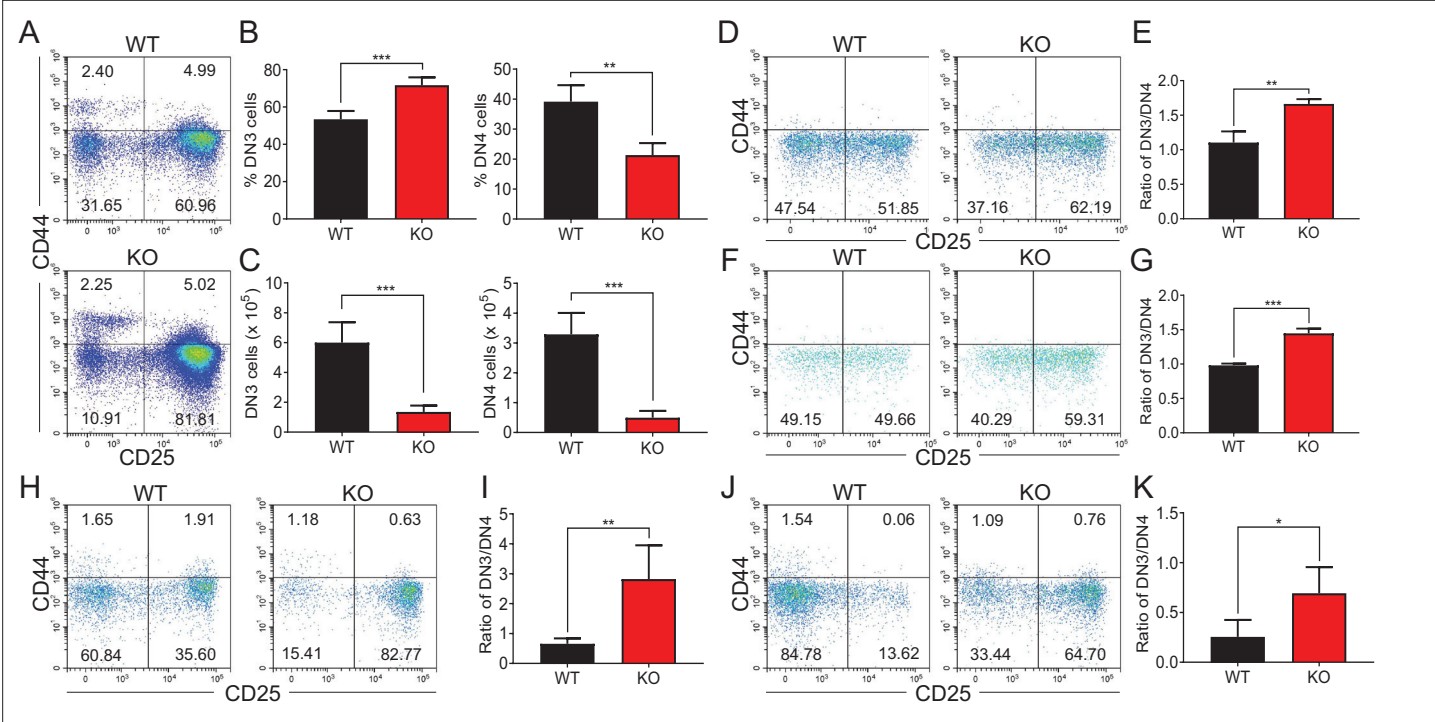

**Figure 3.** *Zfp335*-deficient thymocytes undergo a developmental block during DN3 to DN4 transition. (**A–C**) Thymi were harvested from 6- to 8-week-old WT and KO mice. The different stages of DN thymocytes in WT and KO mice were measured by flow cytometry (*n* = 4). (**A**) Representative FACS plots of DN1 (CD25⁻CD44⁺), DN2 (CD25⁺CD44⁺), DN3 (CD25⁺CD44⁻), and DN4 (CD25⁻CD44⁻) thymocytes. (**B**) The percentages of DN3 and DN4 thymocytes. (**C**) The numbers of DN3 and DN4 thymocytes. (**D–E**) WT and KO DN3 thymocytes were cultured with OP9-DL1 feeder cells in vitro in the presence of IL-7 (1 ng/ml) and Flt3L (5 ng/ml) for 2 days. The expression of CD44 versus CD25 was measured by flow cytometry (*n* = 3). (**D**) Representative FACS plots of DN3 and DN4 thymocytes. (**E**) The ratio of DN3 to DN4 thymocytes 2 days post culture in vitro. (**F–G**) A mixed population of WT and KO DN3 thymocytes at a 1:4 ratio was cocultured with OP9-DL1 feeder cells in vitro in the presence of IL-7 (1 ng/ml) and Flt3L (5 ng/ml) for 2 days. The expression of CD44 versus CD25 was measured by flow cytometry (*n* = 3). (**F**) Representative FACS plots of DN3 and DN4 thymocytes. (**G**) The ratio of DN3 to DN4 thymocytes 2 days post culture in vitro. (**H, I**) Full chimeric mice were generated by transplanting WT (CD45.1⁺) or KO (CD45.2⁺) bone marrow progenitor cells into lethally irradiated (8.5 Gy) WT recipients (CD45.1⁺CD45.2⁺). Six weeks after transplantation, thymi from recipient mice were harvested. The expression of CD44 versus CD25 was measured by flow cytometry (*n* = 4). (**H**) Representative FACS plots of DN3 and DN4 cells. (**I**) The ratio of DN3 to DN4 thymocytes. (**J, K**) Full chimeric mice were generated by transplanting a mixed population of WT (CD45.1⁺) and KO (CD45.2⁺) bone marrow progenitor cells at a 1:4 ratio into lethally irradiated (8.5 Gy) WT recipient mice (CD45.1⁺CD45.2⁺). Six weeks after transplantation, thymi from recipient mice were harvested. The expression of CD44 versus CD25 was measured by flow cytometry (*n* = 4). (**J**) Representative FACS plots of DN3 and DN4 thymocytes. (**K**) The ratio of DN3 to DN4 thymocytes. Results represent three independent experiments. *$p < 0.05$, **$p < 0.01$, and ***$p < 0.001$.

The online version of this article includes the following source data for figure 3:

**Source data 1.** *Figure 3B, C*. The percentages and numbers of DN3 and DN4 thymocytes from WT and KO mice.

cells (*Figure 5—figure supplement 3*). Thus, the reduced iTCRβ expression may be a consequence of protein degradation.

Given that Zfp355 deficiency led to diminished iTCRβ expression in DN4 cells, we next investigated the effect of TCR overexpression on aberrant thymocyte development caused by Zfp335 deficiency. Offspring (*LckCre⁺Zfp335^fl/fl^OT1⁺*) of *LckCre⁺Zfp335^fl/fl^* mice crossed to OT1 transgenic (Tg) mice was generated to constitutively express *Tcra-V2* and *Tcrb-V5* Tg gene (OT1^Tg^ KO). Notably, forced expression of αβTCR successfully restored the decreased percentage of iTCRβ⁺ DN4 cells in the KO mice, despite with little impact on the number of iTCRβ⁺ DN4 cells (*Figure 5G–I*). Importantly, developmental arrest at the DN3 stage in *Zfp335*-deficient mice was fully rescued by OT1 transgene (*Figure 5J–L*). Unfortunately, DN3 and DN4 cells from OT1^Tg^ KO mice still exhibited a similar degree of apoptosis with *Zfp335*-deficient cells (*Figure 5—figure supplement 4*), suggesting Zfp335 affects thymocyte apoptosis in a TCR-independent manner. Consistently, the proportions of DN, DP, CD4, and CD8 were still comparable in KO and OT1^Tg^ KO mice (*Figure 5M, N*; *Figure 5—figure*

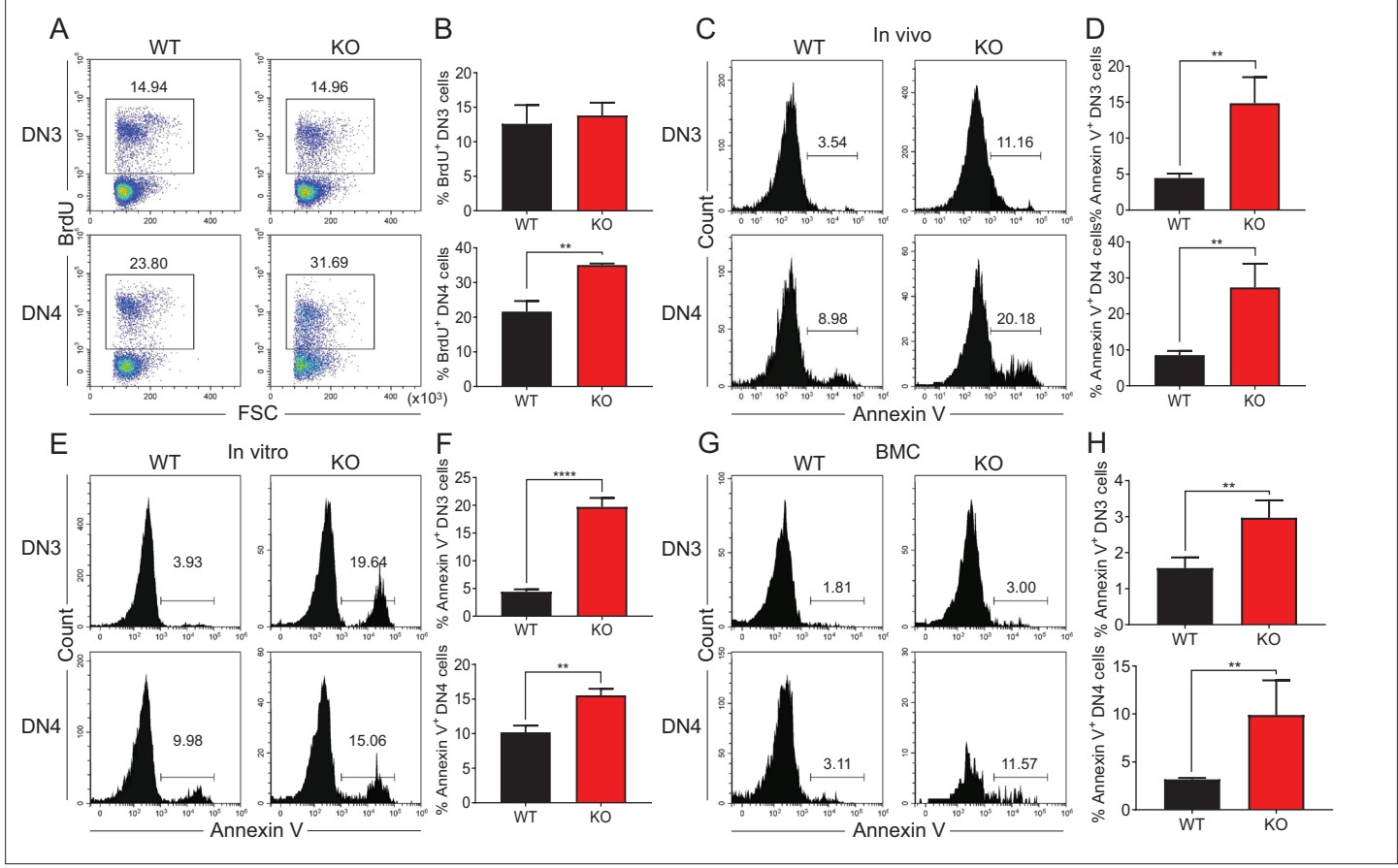

**Figure 4.** Zfp335 deficiency promotes thymocyte apoptosis in vivo and in vitro. (**A, B**) Thymi were harvested from 6- to 8-week-old WT and KO mice. The expression of BrdU in DN3 and DN4 thymocytes from WT and KO thymi was measured by flow cytometry ($n = 3$). (**A**) Representative FACS plots of BrdU expression in DN3 and DN4 thymocytes. (**B**) The percentages of BrdU+ thymocytes in DN3 and DN4 cells. (**C, D**) The binding of Annexin V in DN3 and DN4 thymocytes from WT and KO thymi was measured by flow cytometry ($n = 3$). (**C**) Representative FACS plots of Annexin V binding in DN3 and DN4 thymocytes. (**D**) The percentages of Annexin V+ DN3 and DN4 cells. (**E, F**) A mixed population of WT and KO DN3 thymocytes at a 1:4 ratio was cocultured with OP9-DL1 feeder cells in vitro in the presence of IL-7 (1 ng/ml) and Flt3L (5 ng/ml) for 4 days. The binding of Annexin V was measured by flow cytometry ($n = 3$). (**E**) Representative FACS plots of Annexin V+ DN3 and DN4 cells. (**F**) The percentage of Annexin V+ DN3 and DN4 cells. (**G, H**) Full chimeric mice were generated by transplanting a mixed population of WT (CD45.1+) and KO (CD45.2+) bone marrow progenitor cells at a 1:4 ratio into lethally irradiated WT recipient mice (CD45.1+CD45.2+) with 8.5 Gy. Five weeks after transplantation, thymi from recipient mice were harvested. The binding of Annexin V was measured by flow cytometry ($n = 4$). (**G**) Representative FACS plots of Annexin V+ DN3 and DN4 cells. (**H**) The percentage of Annexin V binding in DN3 and DN4 thymocytes. Results represent three independent experiments. **$p < 0.01$ and ****$p < 0.0001$.

The online version of this article includes the following source data and figure supplement(s) for figure 4:

**Source data 1.** *Figure 4B*. The percentages of BrdU+ thymocytes in DN3 and DN4 cells from WT and KO mice.

**Figure supplement 1.** Zfp335 deficiency has no effect on the proliferation of DN3 and DN4 cells.

**Figure supplement 1—source data 1.** *Figure 4—figure supplement 1B*. The percentages of Ki67+ thymocytes in DN3 and DN4 cells from WT and KO mice.

**Figure supplement 2.** Effect of *Zfp335* deletion on DN3a and DN3b Cells.

**Figure supplement 2—source data 1.** *Figure 4—figure supplement 2E, F*. The percentages of Annexin V+ DN3a and DN3b cells from WT and KO mice.

*supplement 5*), demonstrating that the aberrant DP development has not yet been restored by αβTCR overexpression.

## Zfp335 directly targets *Bcl6* and *Rorc* in DN thymocytes

To address the underlying molecular mechanisms of Zfp335-mediated thymocyte development, we performed comprehensive RNA-seq analysis comparing DN4 cells from WT and KO mice. Volcano

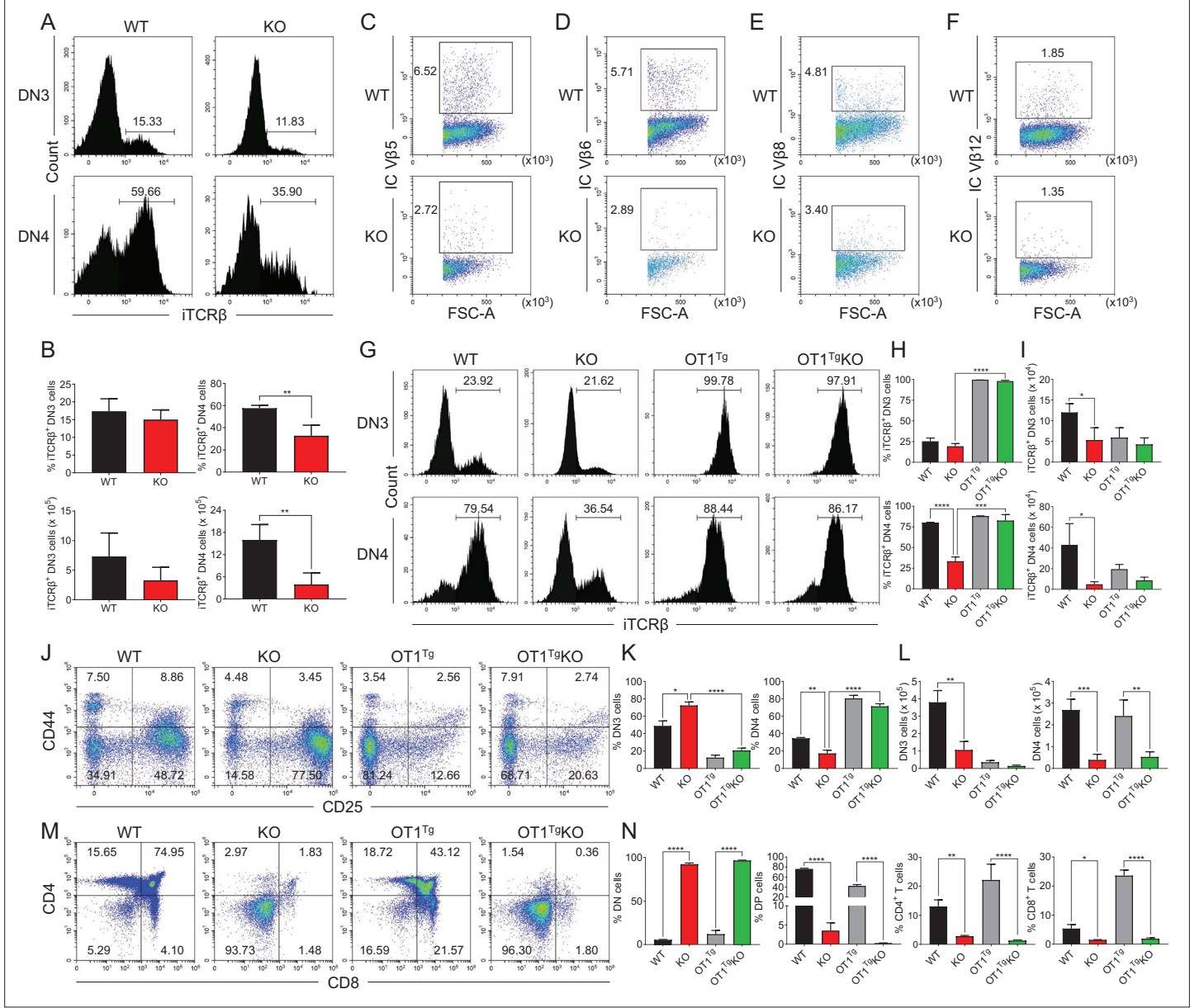

**Figure 5.** OT1 transgenic TCR overexpression rescued Zfp335 deficiency-induced defect during DN3 to DN4 transition. (**A, B**) Thymi were harvested from 6- to 8-week-old WT and KO mice. The expression of iTCRβ in DN3 and DN4 thymocytes was measured by flow cytometry (WT $n = 3$; KO $n = 4$). (**A**) Representative FACS plots of iTCRβ expression in DN3 and DN4 thymocytes. (**B**) Percentages and numbers of iTCRβ+ thymocytes in DN3 and DN4 cells. Representative FACS plots of intercellular TCR Vβ5 (C), Vβ6 (D), Vβ8 (E), and Vβ12 (F) expression in DN4 cells from WT and KO mice. (**G–I**) Thymi from WT, KO, OT1+ (OT1Tg), and OT1+LckCre+Zfp335fl/fl (OT1Tg KO) mice were harvested. The expressions of iTCRβ in DN3 (up) and DN4 (down) thymocytes were measured by flow cytometry (WT $n = 3$; KO $n = 5$; OT1Tg $n = 3$; OT1Tg KO $n = 5$). (**G**) Representative FACS plots of iTCRβ expression in DN3 and DN4 thymocytes. (**H**) The percentage of iTCRβ+ DN3 and DN4 cells in WT, KO, OT1Tg, and OT1Tg KO mice. (**I**) The numbers of iTCRβ+ DN3 and DN4 cells in WT, KO, OT1Tg, and OT1Tg KO mice. (**J–L**) The different stages of DN thymocytes in WT, KO, OT1Tg, and OT1Tg KO mice were measured by flow cytometry (WT $n = 3$; KO $n = 5$; OT1Tg $n = 3$; OT1Tg KO $n = 5$). (**J**) Representative FACS plots of DN3 and DN4 thymocytes. (**K**) The percentages of DN3 and DN4 thymocytes. (**L**) The numbers of DN3 and DN4 thymocytes. (**M, N**) The different stages of thymocyte development in WT, KO, OT1Tg, and OT1Tg KO mice were measured by flow cytometry (WT $n = 3$; KO $n = 5$; OT1Tg $n = 3$; OT1Tg KO $n = 5$). (**M**) Representative FACS plots of thymocytes. (**N**) The percentages of DN, DP, CD4+CD8−, and CD4−CD8+ thymocytes. Results represent three independent experiments. $n = 3$ per group. *$p < 0.05$, **$p < 0.01$, ***$p < 0.001$, and ****$p < 0.0001$.

The online version of this article includes the following source data and figure supplement(s) for figure 5:

**Source data 1.** *Figure 5B*. The percentages and numbers of iTCRβ+ thymocytes in DN3 and DN4 cells from WT and KO mice.

**Figure supplement 1.** Effect of *Zfp335* deletion on pre-TCR complex expression in DN3 and DN4 cells.

*Figure 5 continued on next page*

*Figure 5 continued*

**Figure supplement 2.** Intercellular CD3 (iCD3) expression in *Zfp335*-deficient DN3 and DN4 thymocytes.

**Figure supplement 2—source data 1.** *Figure 5—figure supplement 2B*. Intercellular CD3 expression in Zfp335-deficient DN3 and DN4 thymocytes.

**Figure supplement 3.** Effect of *Zfp335* deletion on TCRβ rearrangement in DN3 thymocytes.

**Figure supplement 3—source data 1.** The gel of CD14 gene.

**Figure supplement 3—source data 2.** The gel of TCR Vb5.

**Figure supplement 3—source data 3.** The gel of TCR Vb8.

**Figure supplement 4.** TCRβ overexpression could not rescue the increased DN cell apoptosis caused by Zfp335 deficiency.

**Figure supplement 4—source data 1.** *Figure 5—figure supplement 4B, C*. The percentages of Annexin V+ DN3 and DN4 cells in WT, KO, OT1Tg, and OT1Tg KO mice.

**Figure supplement 5.** OT1Tg failed to rescue thymocyte numbers.

**Figure supplement 5—source data 1.** *Figure 5—figure supplement 5*. The numbers of DN, DP, CD4+, and CD8+ thymocytes in WT, KO, OT1Tg, and OT1Tg KO mice.

plot showed that *Zfp335*-deficient DN4 cells had a total 566 downregulated and 899 upregulated genes (fold change >1.25, $p < 0.05$) (*Figure 6A*, *Supplementary file 1*). Gene ontology (GO) analysis highlighted a large fraction of genes downregulated in KO group belonging to lymphocyte differentiation and apoptotic pathways (*Figure 6B*). Prominently downregulated genes associated with lymphocyte differentiation and apoptosis were summarized in the heatmap (*Figure 6C*). To further determine genes directly regulated by Zfp335, we analyzed Zfp335 by chromatin immunoprecipitation followed by deep sequencing (ChIP-seq). We screened a total of 2797 Zfp335-binding sites (*Supplementary file 2*) and the prevalence of binding peaks across genomic regions was displayed as a pie chart (*Figure 6—figure supplement 1*). To identify the Zfp335-targeting candidates, 119 profoundly downregulated genes were further selected out by a cutoff of twofold change, and 22 genes were overlapped with Zfp335-targeting genes from ChIP-seq data (*Figure 6—figure supplement 2A*, *Supplementary file 3*). Among these genes, top 10 genes were listed based upon their expression level from RNA-seq result (*Figure 6—figure supplement 2B*). qPCR analysis was performed to confirm their downregulation in KO DN4 cells (*Figure 6—figure supplement 2C*). Next, we focused on genes related to lymphocyte differentiation and apoptosis. In line with RNA-seq results (*Figure 6C*), qPCR analysis verified that *Bcl6* and *Rorc* were significantly downregulated in KO DN4 cells (*Figure 6D*). Importantly, Zfp335 directly targeted the promoter regions of *Bcl6* and *Rorc* in ChIP-seq analysis (*Figure 6E*), which were further verified by luciferase assay (*Figure 6F*). Taken together, in depth genomic analysis of DN thymocytes supports that Zfp335 directly regulates the transcription of *Bcl6* and *Rorc*.

## Defects in thymocyte development caused by Zfp335 deficiency can be rescued by *Bcl6* and *Rorc*

To determine whether *Bcl6* and *Rorc* participate in the regulation of thymocyte development downstream of Zfp335, overexpression of *Bcl6* and *Rorc* was conducted in vitro in the thymocyte development model (*Figure 7A*). DN3 cells from KO mice were cocultured with OP9-DL1 cells and transduced with retrovirus encoding *Mock*-GFP, *Zfp335*-GFP, *Bcl6*-GFP, or *Rorc*-GFP. After 3.5 days, overexpression of *Bcl6* and *Rorc* resulted in a substantial DP generation, particularly in the *Bcl6* group, resulting in a similar DP proportion to that in *Zfp335*-overexpressing cells (*Figure 7B, C*). Of note, overexpression of *Psmg2*, *Dctn1*, *Ankle2*, *Cep76*, *Fgf13*, and *Ddx31* identified in the qPCR results (*Figure 6—figure supplement 2B, C*) in KO DN3 cells did not restore DP generation in vitro (*Figure 7—figure supplement 1*). Importantly, enhanced expression of *Bcl6* in DN3 cells rescued DN thymocyte apoptosis (*Figure 7D, E*), while overexpression of both *Bcl6* and *Rorc* rescued DP thymocytes from enhanced apoptosis (*Figure 7F, G*). p53, negatively regulated by Bcl6, is involved in lymphocyte apoptosis (*Haks et al., 1999*; *Guidos et al., 1996*; *Mombaerts et al., 1995*). Thus, we crossed *Zfp335* KO strain with *Trp53* KO strain to obtain double knockout mice, in which *Trp53* deletion resulted in a partial recovery of DP percentage, but not cell number (*Figure 7—figure supplement 2*), further supporting the role of *Bcl6* in *Zfp335*-controlled thymocyte survival. Together, these data demonstrate that Zfp335 controls DN thymocyte survival through direct regulation of *Bcl6* and *Rorc* expression.

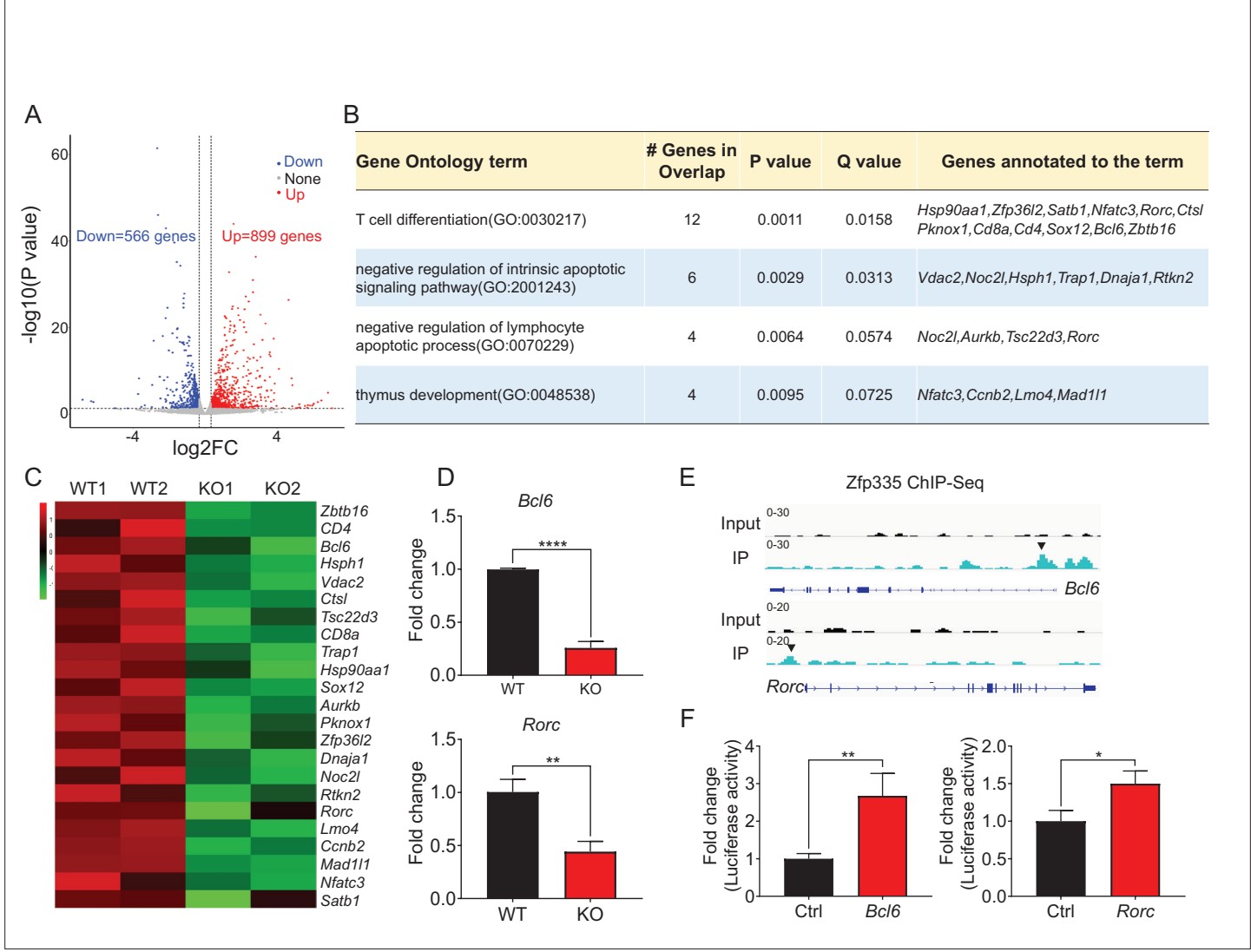

**Figure 6.** Zfp335 downstream target analysis in DN4 thymocytes. (**A**) Volcano plot depicting log$_2$ (fold change) (x-axis) and −log$_{10}$ (p value) (y-axis) for differentially expressed genes (FC >1.25, p < 0.05) in DN4 thymocytes sorted from WT and KO mice; upregulated (red) and downregulated (blue). n = 2 per group. (**B**) Gene ontology (GO) analysis of genes that downregulated in *Zfp335*-deficient DN4 thymocytes, showing the GO terms related to lymphocyte differentiation and apoptosis (left), the number of genes overlapped with database from the indicated terms (middle left column), p values (middle column) and Q values (middle right column) and genes annotated to the indicated term (right). (**C**) Heatmap of representative genes related to lymphocyte differentiation and apoptosis. The scale ranges from minimum (green boxes) to medium (black boxes) to maximum (red boxes) relative expression. (**D**) The mRNA level of *Bcl6* (top) and *Rorc* (bottom) in DN4 thymocytes from WT and KO mice (n = 3). (**E**) ChIP-seq analysis for binding of Zfp335 to the *Bcl6* and *Rorc* loci in wild-type DN4 cells. (**F**) Luciferase assay for the binding of different domains of Zfp335 to the promoter regions of *Bcl6* and *Rorc*. The pGL4.16 plasmid was transfected into 293T cells together with MSCV vector carrying different domains (n = 3). Data represent three independent experiments. *p < 0.05, **p < 0.01, and ****p < 0.0001.

The online version of this article includes the following figure supplement(s) for figure 6:

**Figure supplement 1.** ChIP-seq analysis of Zfp335-binding sites.

**Figure supplement 2.** The intersection analysis of Zfp335 by RNA- and ChIP-seq.

## Discussion

In this study, we reveal that Zfp335 is essential for thymocyte development, particularly during DN to DP transition. Zfp335 deficiency in T cells led to a significant loss of DP, CD4 SP, and CD8 SP cells while an accumulation of DN3 cells. Mechanistically, the developmental blockade is attributed to both impaired pre-TCR signal and increased susceptibility to apoptosis. Serving as a transcription factor,

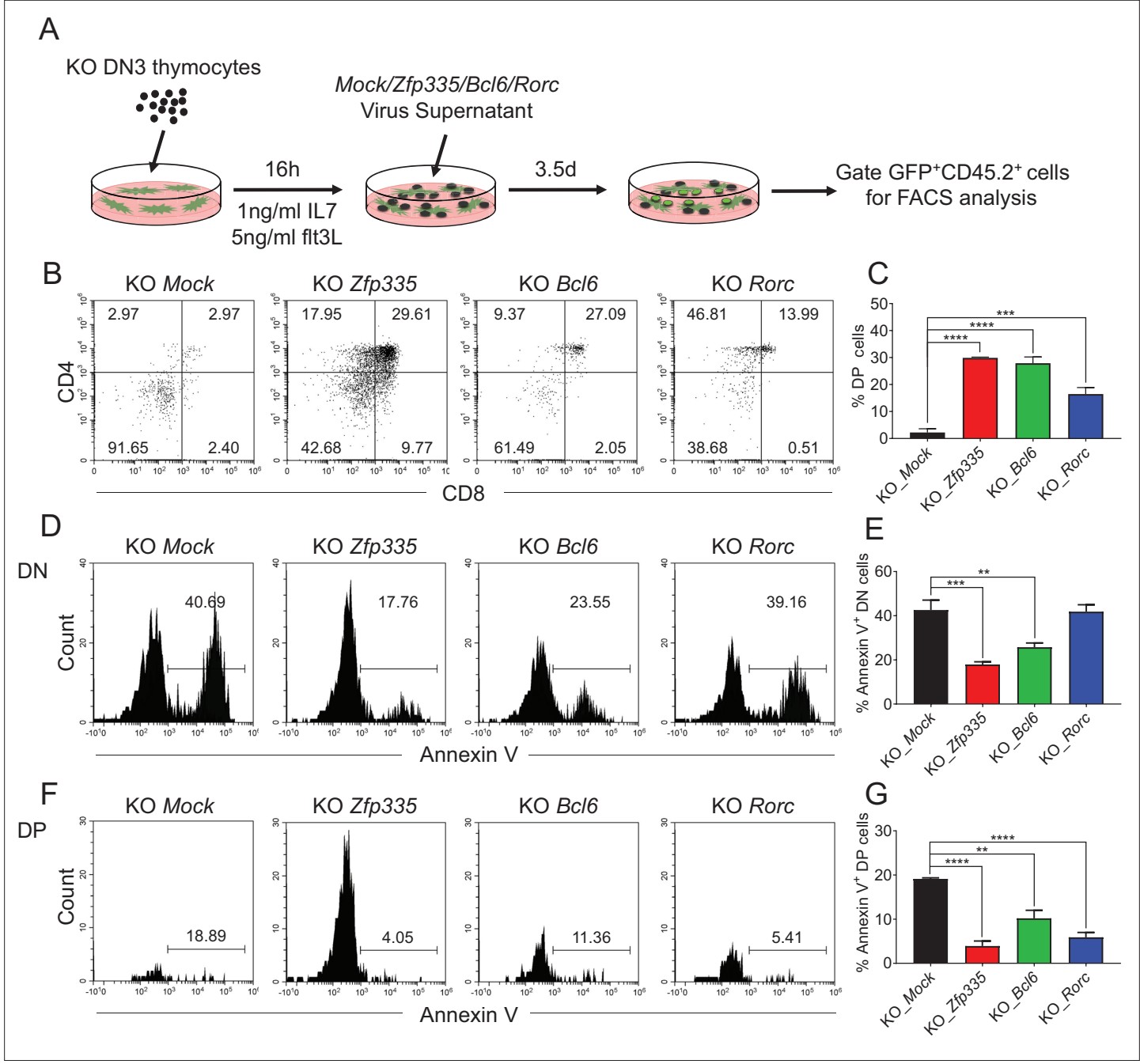

**Figure 7.** Identification of *Bcl6* and *Rorc* as functional targets of Zfp335 for regulating thymocyte development. (**A**) Schematic overview of the in vitro gene overexpression in KO DN3 thymocytes and following T-cell differentiation in OP9-DL1 coculture system. (**B, C**) *Zfp335*-deficient DN3 thymocytes (KO) were cultured with OP9-DL1 feeder cells in vitro, then transduced with either *Mock*, *Zfp335*-, *Bcl6*-, or *Rorc*-overexpressing vector for 3.5 days. The different stages of thymocyte development from GFP-positive cells were measured by flow cytometry (*n* = 3). (**B**) Representative FACS plots of DN and DP thymocytes from the indicated groups. (**C**) The percentage of DP thymocytes from GFP-positive cells. (**D–G**) KO DN3 thymocytes were cultured with OP9-DL1 feeder cells in vitro, then transduced with either *Mock*, *Zfp335*-, *Bcl6*-, or *Rorc*-overexpressing vector for 3.5 days. The expressions of Annexin V in DN and DP thymocytes were measured by flow cytometry (*n* = 3). (**D**) Representative FACS plots of Annexin V⁺ DN thymocytes from the indicated groups. (**E**) The percentage of Annexin V⁺ DN thymocytes from GFP-positive cells. (**F**) Representative FACS plots of Annexin V⁺ DP thymocytes from the indicated groups. (**G**) The percentage of Annexin V⁺ DP thymocytes from GFP-positive cells. Results shown represent three independent experiments. **p < 0.01, ***p < 0.001, and ****p < 0.0001.

The online version of this article includes the following source data and figure supplement(s) for figure 7:

**Source data 1.** *Figure 7C*. The percentages of DP cells differentiated from KO DN3 thymocytes transduced with Mock, or Zfp335, Bcl6, and Rorc genes.

*Figure 7 continued on next page*

*Figure 7 continued*

**Figure supplement 1.** Identification of the top 10 overlapped genes in KO DN3 thymocytes for the regulation of thymocyte development.

**Figure supplement 2.** Examination of *Trp53* as the target of Zfp335 during the regulation of thymocyte development.

**Figure supplement 2—source data 1.** *Figure 7—figure supplement 2B, C*. The percentages and numbers of DN, DP, CD4[+], and CD8[+] thymocytes from WT, KO, and Trp53DKO mice.

**Figure supplement 3.** Intracellular TCRβ (iTCRβ) expression in DN4 cells infected by *Zfp335*, *Bcl6*, and *Rorc* overexpression retrovirus.

**Figure supplement 3—source data 1.** *Figure 7—figure supplement 3*. iTCRβ expression in DN4 cells infected by Zfp335, Bcl6, and Rorc overexpression retrovirus.

Zfp335 directly promotes *Bcl6* and *Rorc* expression in DN thymocytes to ensure their survival during early development.

Zfp335 was previously demonstrated to be crucial for early embryonic development as homozygous deletion of this gene resulted in neonatal death (*Garapaty et al., 2009*). Conditional knockout of *Zfp335* in neural system led to severely reduced cortical size and impaired neurogenesis. Mechanistically, Zfp335 was required for neural progenitor cell self-renewal and proliferation, and neuronal differentiation (*Yang et al., 2012*) and neuronal morphogenesis (*Zhao et al., 2015*). Besides, deficiency of naive T cells in mice carrying a hypomorph allele of *Zfp335* (*Zfp335^bloto*) uncovered its role in immune system (*Han et al., 2014*). So far, there is still very limited information about the functions of Zfp335 in other aspects of immune system. Here, we found that Zfp335 is absolutely required for multiple steps of early T-cell development. Of note, it will be worth investigating whether and how Zfp335 is involved in the regulation of mature T-cell differentiation and functions under static and immunized conditions in future.

We have shown that Zfp335 expression was upregulated specifically in DN3 thymocytes and significantly decreased in the subsequent stages, suggesting a critical role at the DN3 stage. Of note, loss of Zfp335 led to a dramatic reduction in both thymus size and thymocyte number. The accumulation of DN3 cells results from an intrinsic mechanism that hinders the transition from DN3 to DN4 stage, leading to the reduction of DP thymocytes and mature T cells in the periphery. These data are in line with another recent study reporting that *Zfp335* mutation led to a reduction in peripheral T cells as a result of defective naive T cells and SP thymocytes (*Han et al., 2014*). However, given the intact thymic selection with *Zfp335* mutation, the report was inconsistent with our observation of decreased β-selection with Zfp335 deficiency. The discrepancy is likely due to the different approaches used to disrupt Zfp335 function since a single-nucleotide missense mutation of Zfp335 may affect its function differently. Nevertheless, by knocking out the entire Zfp335 protein, we provide evidence that Zfp335 is indispensable for early thymocyte development.

Thymocyte β-selection is a critical developmental checkpoint allowing for the progression from DN3 to DN4 stage and the maintenance of DP cell numbers, which is primarily dependent on TCRβ and pre-TCR signals constituted with a functional iTCRβ paired with a pTα chain (*Yamasaki and Saito, 2007*). Pre-TCR signaling regulates thymocytes differentiation, proliferation, and survival in the full developmental process (*Koch and Radtke, 2011*). In addition, there are reports that other pathways or transcription factors coupled with or independent of conventional pre-TCR signaling are found to play important roles in the process (*Ciofani et al., 2004*; *Pajerowski et al., 2009*; *Zikmund et al., 2019*). While *Zfp335*-deficient DN4 cells exhibited no defects in the rearrangement of TCRβ chain genes and pTα gene expression, our results clearly demonstrated that Zfp335 deficiency markedly impaired iTCRβ expression and led to an unbiased reduction of the majority of Vβ genes in DN4 populations. Future studies will investigate the mechanisms how Zfp335 regulates iTCRβ expression. Importantly, forced iTCRβ expression in DN3 and DN4 cells by transduction of OT1-TCR completely rescued the developmental impairment during the DN3-DN4 transition in *Zfp335*-deficient mice despite the failure to rescue the DN3, DN4, and DP population size. This suggests that Zfp335 controls the DN3–DN4 transition dependent on pre-TCR signals, but other mechanisms may also regulate the DP population size.

The large population of DP thymocytes is maintained by both cell proliferation and survival mechanisms. *Zfp335*-deficient DN3 and DN4 cells showed slightly higher or unchanged incorporation of BrdU, suggesting that cell proliferation was not affected. However, our data revealed a significant increase in apoptosis in *Zfp335*-deficient DN3 and DN4 thymocytes. Transcriptomic analysis (RNA-seq

and qPCR) unveiled the downregulation of *Bcl6* and *Rorc* signaling which are critically involved in thymocyte apoptosis (*Solanki et al., 2020*; *Xi et al., 2006*). Indeed, we have demonstrated that Zfp335, a transcription factor, direct bound to the promoter regions of *Bcl6* and *Rorc* genes. More importantly, enhanced expression of *Bcl6* and *Rorc* could improve thymocyte survival and substantially restore the DP thymocyte population. Trp53 deletion resulted in a partial recovery of DP cells, further supporting the role of *Bcl6* in *Zfp335*-controlled thymocyte survival. Of note, Zfp335 may also control thymocyte survival through directly regulating other targets.

In our study, Zfp335 is indispensable for thymocyte β-selection given that T-cell-specific deficiency in Zfp335 leads to impaired iTCRβ expression, blockade of thymocytes at DN stage as well as a substantial DN cell apoptosis. Though enhanced expression of TCRβ restores the developmental defect during DN3 to DN4 transition, it had little impact on the population size of DN3, DN4, and DP cells, suggesting the regulation of Zfp335 on DN cell apoptosis through mechanisms more than β-selection. Indeed, we provided the evidence that Zfp335-controlled DN cell survival through regulating *Bcl6* and *Rorc* expression. Moreover, Zfp335 regulates TCRβ expression independent on *Bcl6* and *Rorc* since overexpression of neither *Bcl6* or *Rorc* in *Zfp335*-deficient DN3 thymocytes could restore the decreased iTCRβ expression (*Figure 7—figure supplement 3*).

Several key factors have been shown to involve in the regulation of T-cell developmental process from DN3 to DP stages. Notch signaling is required for early T-cell commitment and β-selection (*Radtke et al., 2010*; *Ciofani and Zúñiga-Pflücker, 2005*; *Hosokawa and Rothenberg, 2018*), which is subsequently weaken by Bcl6 repression for the differentiation from DN to DP stage development (*Solanki et al., 2020*). Consistently, our results found that Zfp335 could directly target *Bcl6*, and Zfp335 deficiency led to decreased expression of *Bcl6* and upregulation of Notch target genes such as *Dtx1* and *Notch1* (data not shown), which collectively contributing to the developmental block from DN to DP stage. In addition, Tcf1 plays a vital role in T-cell lineage commitment since *Tcf1*[−/−] DN3 thymocytes failed to progress to DN4 and subsequent DP stage through regulating *Bcl11b* and *Gata3* expression (*Garcia-Perez et al., 2020*). Bcl11b and Gata3 also differentially regulate the differentiation and survival of thymocytes at DN3, DN4, and subsequent ISP stages via TCR or/and survival signals (*Inoue et al., 2006*; *Pai et al., 2003*). However, we did not detect reduced expression of *Tcf1*, *Bcl11b*, or *Gata3* in *Zfp335*-deficient DN4 cells, suggesting that Zfp335 regulates DN3 to DP thymocyte development independent on these molecules.

In conclusion, our study reveals that the C2H2 zinc finger protein Zfp335 plays a novel and crucial role during thymocyte development, specifically during the transition from DN to DP stage. Mechanistically, Zfp335 promotes *Bcl6* and *Rorc* signaling to prevent thymocytes apoptosis and ensure the survival and differentiation of thymocytes. Collectively, we provide evidence that Zfp335 is essential for thymocyte development through both pre-TCR-dependent and -independent mechanisms.

# Materials and methods

## Key resources table

| Reagent type (species) or resource | Designation | Source or reference | Identifiers | Additional information |
|---|---|---|---|---|
| Genetic reagent (*M. musculus*) | C57BL/6J background | Jackson Laboratory | Stock No. 000664 | |
| Genetic reagent (*M. musculus*) | C57BL/6-*Zfp335tm1Caw* | Jackson Laboratory | Stock No. 022413 | |
| Genetic reagent (*M. musculus*) | C57BL/6-Tg(*TcraTcrb*)1,100Mjb/J | Jackson Laboratory | Stock No. 003831 | Common Name: OT-1 |
| Genetic reagent (*M. musculus*) | B6.Cg-Tg(Lck-cre)548Jxm/J | Jackson Laboratory | Stock No. 003802 | |
| Genetic reagent (*M. musculus*) | C3Ou.129S2(B6)-Trp53tm1Tyj/J | Jackson Laboratory | Stock No. 002547 | |
| Antibody | anti-mouse CD4 APC/Cyanine7 (Rat monoclonal) | Biolegend | Cat# 100414; RRID:AB_312699 | cell surface staining 1:400 |

*Continued on next page*

*Continued*

| Reagent type (species) or resource | Designation | Source or reference | Identifiers | Additional information |
|---|---|---|---|---|
| Antibody | anti-mouse CD8a PE (Rat monoclonal) | Biolegend | Cat# 100708; RRID:AB_312747 | cell surface staining 1:400 |
| Antibody | anti-mouse CD8a Pacific Blue (Rat monoclonal) | Biolegend | Cat# 100725; RRID:AB_493425 | cell surface staining 1:400 |
| Antibody | anti-mouse CD8a PE/Cyanine7 (Rat monoclonal) | Biolegend | Cat# 100722; RRID:AB_312761 | cell surface staining 1:400 |
| Antibody | anti-mouse TCR Vβ5.1, 5.2 PE (Mouse monoclonal) | Biolegend | Cat# 139504, RRID:AB_10613279 | cell surface staining 1:400 |
| Antibody | anti-mouse TCR Vβ6 PE (Rat monoclonal) | Biolegend | Cat# 140004; RRID:AB_10643583 | cell surface staining 1:400 |
| Antibody | anti-mouse TCR Vβ8.1, 8.2 PE (Mouse monoclonal) | Biolegend | Cat# 140104; RRID:AB_10639942 | cell surface staining 1:400 |
| Antibody | anti-mouse TCR Vβ12 PE (Mouse monoclonal) | Biolegend | Cat# 139704; RRID:AB_10639729 | cell surface staining 1:400 |
| Antibody | anti-mouse TCR β chain APC/Cyanine7 (Armenian Hamster monoclonal) | Biolegend | Cat# 109220; RRID:AB_893624 | cell surface staining 1:400 |
| Antibody | anti-mouse TCR β chain PE/Cyanine5 (Armenian Hamster monoclonal) | Biolegend | Cat# 109210; RRID:AB_313433 | cell surface staining 1:400 |
| Antibody | anti-mouse TCR β chain PE/Cyanine7 (Armenian Hamster monoclonal) | Biolegend | Cat# 109222; RRID:AB_893625 | cell surface staining 1:400 |
| Antibody | anti-mouse TCR γ/δ FITC (Armenian Hamster monoclonal) | Biolegend | Cat# 118106; RRID:AB_313830 | cell surface staining 1:400 |
| Antibody | anti-mouse TCR γ/δ PerCP/Cyaninne5.5 (Armenian Hamster monoclonal) | Biolegend | Cat# 118118; RRID:AB_10612756 | cell surface staining 1:400 |
| Antibody | anti-mouse TCR γ/δ APC (Armenian Hamster monoclonal) | Biolegend | Cat# 118116; RRID:AB_1731813 | cell surface staining 1:400 |
| Antibody | anti-mouse/human CD44 PE/Cyanine7 (Rat monoclonal) | Biolegend | Cat# 103030; RRID:AB_830787 | cell surface staining 1:400 |
| Antibody | anti-mouse CD25 PE (Rat monoclonal) | Biolegend | Cat# 102008; RRID:AB_312857 | cell surface staining 1:400 |
| Antibody | anti-mouse CD25 PE/Cyanine5 (Rat monoclonal) | Biolegend | Cat# 102010; RRID:AB_312859 | cell surface staining 1:400 |
| Antibody | anti-mouse CD4 FITC (Rat monoclonal) | Biolegend | Cat# 100510; RRID:AB_312713 | cell surface staining 1:400 |
| Antibody | anti-mouse CD8a FITC (Rat monoclonal) | Biolegend | Cat# 100706; RRID:AB_312745 | cell surface staining 1:400 |
| Antibody | anti-mouse TCR β chain FITC (Armenian Hamster monoclonal) | Biolegend | Cat# 109206; RRID:AB_313429 | cell surface staining 1:400 |
| Antibody | anti-mouse NK-1.1 FITC (Mouse monoclonal) | Biolegend | Cat# 108706; RRID:AB_313393 | cell surface staining 1:400 |
| Antibody | anti-mouse CD19 FITC (Rat monoclonal) | Biolegend | Cat# 115506; RRID:AB_313641 | cell surface staining 1:400 |
| Antibody | anti-mouse CD11b FITC (Rat monoclonal) | Biolegend | Cat# 101206; RRID:AB_312789 | cell surface staining 1:400 |
| Antibody | anti-mouse CD11c FITC (Armenian Hamster monoclonal) | Biolegend | Cat# 117306; RRID:AB_313775 | cell surface staining 1:400 |
| Antibody | anti-mouse TER-119/Erythroid Cells FITC (Rat monoclonal) | Biolegend | Cat# 116206; RRID:AB_313707 | |

*Continued on next page*

Continued

| Reagent type (species) or resource | Designation | Source or reference | Identifiers | Additional information |
|---|---|---|---|---|
| Antibody | anti-mouse TCR β chain Pacific Blue (Armenian Hamster monoclonal) | Biolegend | Cat# 109226; RRID:AB_1027649 | |
| Antibody | anti-mouse CD45.1 APC/Cy7 (Mouse monoclonal) | Biolegend | Cat# 110716; RRID:AB_313505 | cell surface staining 1:400 |
| Antibody | anti-mouse CD45.2 PE (Mouse monoclonal) | Biolegend | Cat# 109808; RRID:AB_313445 | |
| Antibody | anti-mouse CD45.2 APC (Mouse monoclonal) | Biolegend | Cat# 109814; RRID:AB_389211 | |
| Antibody | anti-mouse CD3 Pacific Blue (Armenian Hamster monoclonal) | Biolegend | Cat# 640918; RRID:AB_493645 | |
| Antibody | anti-mouse CD27 FITC (Armenian Hamster monoclonal) | Biolegend | Cat# 124208; RRID:AB_1236466 | |
| Antibody | anti-BrdU FITC (Mouse monoclonal) | Biolegend | Cat# 364104; RRID:AB_2564481 | |
| Antibody | anti-BrdU FITC (3D4) | Biolegend | Cat# 364104; RRID:AB_2564481 | |
| Antibody | anti-mouse Ki-67 PE (16A8) | Biolegend | Cat# 652404; RRID:AB_2561525 | |
| Antibody | Anti-Zfp335 antibody | Novus | Cat# NB100-2579 | |
| Peptide, recombinant protein | Annexin V Pacific Blue | Biolegend | Cat# 640918; RRID:AB_1279044 | |
| Peptide, recombinant protein | DNaseI | Solarbio | Cat# D8071 | |
| Commercial assay or kit | Fixation Buffer | Biolegend | Cat# 420,801 | |
| Commercial assay or kit | Intracellular Staining Permeabilization Wash Buffer (10×) | Biolegend | Cat# 421,002 | |
| Commercial assay or kit | MojoSort Streptavidin Nanobeads | Biolegend | Cat# 480,016 | |
| Commercial assay or kit | Fixation/Permeabilization Solution Kit with BD GolgiPlug | BD biosciences | Cat# 555,028 | |
| Commercial assay or kit | One-Day Chromatin Immunoprecipitation Kits | MILLIPORE | Cat# 17-10085 | |
| Commercial assay or kit | Quick-RNA MicroPrep Kit | QIAGEN | Cat# R1051 | |
| Commercial assay or kit | Fixation Buffer | Biolegend | Cat# 420,801 | |
| Software, algorithm | FlowJo software v10.7 | FlowJo LLC | https://www.flowjo.com/; RRID:SCR_008520 | |
| Software, algorithm | GSEA | Broad Institute | https://www.broadinstitute.org/gsea; RRID:SCR_003199 | |
| Software, algorithm | Prism8 (v8.1.0) | GraphPad Software | https://www.graphpad.com/ | |
| Software, algorithm | RStudio | RStudio | https://rstudio.com/; RRID: SCR_000432 | |

## Mice

*Zfp335*[f/f], *LckCre*, *Trp53*[DKO], and *OT-1* strains were purchased from The Jackson Laboratory (Bar Harbor, ME, USA). *Lck*Cre mice were crossed with *Zfp335*[f/f] mice to generate *Lck*Cre[+]*Zfp335*[f/f] (KO) mice and *Lck*Cre[+]*Zfp335*[+/+] (WT) mice. Mice aged 6–8 weeks were used for analyses in the study. All mice were housed in specific-pathogen-free conditions by the Xi'an Jiaotong University Division of Laboratory Animal Research. All animal procedures were approved by the Institutional Animal Care and Use Committee of Xi'an Jiaotong University.

## Antibodies and reagents

The following antibodies and kits were purchased from Biolegend (San Diego, CA, USA): APC/Cy7 anti-CD4 (clone GK1.5), PE anti-CD8α (clone 53–6.7), Pacific Blue anti-CD8α (clone 53–6.7), PE/Cyanine7 anti-CD8α (clone 53–6.7), PE anti-TCRVβ5.1, 5.2 (clone MR9-4), PE anti-TCRVβ6 (clone RR4-7), PE anti-TCRVβ8.1, 8.2 (clone MR5-2), PE anti-TCRVβ12 (clone MR11-1), APC/Cyanine7 anti-TCRβ (clone H57-597), PE/Cyanine5 anti-TCRβ (clone H57-597), PE/Cy7 anti-TCRβ (clone H57-597), FITC

anti-TCRγδ (clone GL3), PE/Cyanine5 anti-TCRγδ (clone GL3), APC anti-TCRγδ (clone GL3), PE/Cy7 anti-CD44 (clone IM7), PE anti-CD25 (clone PC61), PE/Cyanine5 anti-CD25 (clone PC61), FITC anti-CD4 (clone GK1.5), FITC anti-CD8 (clone 53-6.7), FITC anti-TCRβ (clone H57-597), FITC anti-NK1.1 (clone PK136), FITC anti-CD19 (clone 6D5), FITC anti-CD11b (clone M1/70), FITC anti-CD11c (clone N418), FITC anti-TER-119/Erythroid Cells (TER-119), Pacific Blue anti-TCRβ (clone H57-597), APC/Cy7 anti-CD45.1 (clone A20), PE anti-CD45.2 (clone 104), APC anti-CD45.2 (clone 104), Pacific Blue anti-Annexin V (Cat # 640918), Pacific Blue anti-CD3 (clone 17A2), FITC anti-CD27 (clone LG.3A10), FITC anti-BrdU (clone Bu20a), and the Fixation/Permeabilization Solution Kit (Cat # 554722). PE anti-Ki67 monoclonal antibody (clone SolA15) was purchased from eBioscience (San Diego, CA, USA). Quick-RNA Microprep Kit (Cat # R1051) was obtained from Zymo Research (Irvine, CA, USA).

## FACS analysis and Sorting

Lymphocytes from *Lck*Cre[+]*Zfp335*[+/+] and *Lck*Cre[+]*Zfp335*[fl/fl] mice were isolated. For surface staining, single cell suspension was prepared. A total of $1 \times 10^6$ cells were stained in the dark at 4°C for 30 min with indicated antibodies. The analysis was performed on a CytoFLEX flow cytometer (Beckman Coulter; Brea, CA, USA). DN3 (Lin[−]CD25[+]CD44[−]) and DN4 (Lin[−]CD25[−]CD44[−]) cells were collected by BD FACSAria Ⅱ cell sorter (BD Biosciences, San Jose, CA, USA). FACS data were recorded and analysed using CytExpert software (Version 2.3.0.84; Beckman Coulter; Indianapolis, IN, USA).

## Intracellular staining

DN and DP thymocytes were phenotyped using a combination of surface antibodies against lineage markers (CD4, CD8α, TCRβ, TCRγδ, NK1.1, CD19, CD11b, CD11c, and Ter119), together with CD44 and CD25 antibodies. For intracellular cytokine staining, the thymocytes were fixed and permeabilized using a Fixation/Permeabilization Solution Kit (Biolegend), followed by staining using indicated antibodies. The cells were analyzed on a CytoFLEX flow cytometer (Beckman Coulter).

## Quantitative RT-PCR

Cell lysis was performed with RNA extraction and cDNA synthesis using Quick-RNA Microprep Kit (Zymo Research) and ReverTra Ace qPCR RT Master Mix Kit (TOYOBO), respectively. The qRT-PCR reactions were carried out using StepOnePlus Real-Time PCR Systems (ABI) with SYBR mixture (Genstar) to determine relative gene expression. The sequences for the primers are list in *Supplementary file 4*.

## Bone marrow transplantation

Lineage-negative BM cells from CD45.1[+] mice and *Lck*Cre[+]*Zfp335*[fl/fl] mice (CD45.2[+]) were sorted, and mixed at a 1:4 ratio and cotransferred into lethally irradiated (8.5 Gy) recipient mice (CD45.1[+]CD45.2[+]). Six weeks after bone marrow transplantation, thymocytes from recipient mice were harvested for FACS analysis.

## In vitro OP9-DL1 cell coculture

Both Lin[−]CD25[+]CD44[−] DN3 cells and Lin[−]CD25[−]CD44[−] DN4 cells were sorted from the thymi of *Lck*Cre[+]*Zfp335*[+/+] and *Lck*Cre[+]*Zfp335*[fl/fl] mice and cocultured with OP9-DL1 feeder cells in α-MEM medium in the presence of IL-7 (1 ng/ml, PeproTech) and Flt3-L (5 ng/ml, PeproTech). On days 2 and 4, total cells were collected and stained with indicated antibodies for FACS analysis.

## Retroviral transduction of DN3 thymocytes

Retroviruses were produced from BOSC cells transfected with *Mock*-GFP, *Zfp335*-GFP, *Bcl6*-GFP, and *Rorc*-GFP retroviral plasmids. For retroviral transduction, Lin[−]CD25[+]CD44[−] DN3 thymocytes were sorted by FACSAria flow cytometry (BD) and cocultured with OP9-DL1 feeder cells overnight in the presence of 1 ng/ml IL-7 and 5 ng/ml Flt3L. Retroviral infection was performed 16 hr later by centrifugation (2500 rpm for 90 min at 37°C) in the presence of retroviral supernatants and 8 µg/ml polybrene. After spinning, supernatants were replaced by α-MEM medium with 10% FCS supplemented with 1 ng/ml IL-7 and 5 ng/ml Flt3L. 3.5 days later, GFP[+] cells were examined using flow cytometry analysis.

## Luciferase assay

To assess whether Zfp335 regulates *Bcl6* and *Rorc* by directly binding to their promoter regions, the DNA fragments were cloned into the pGL4.16 (luc2CP/Hgro) vector (Promega) which contains the luciferase reporter gene luc2CP. The pGL4.16 plasmid, control vector pGL4.74 (hRluc/TK) encoding the luciferase reporter gene hRluc (*Renilla reniformis*), along with plasmids expressing candidate genes were transfected separately into 293T cell line (ATCC). Forty-eight hours post-transfection, the luciferase activity of both Firefly and Renilla luciferase was measured using a Dual-Luciferase Reporter kit (Promega) on SYNERGY Neo2 multimode reader (BioTek).

## RNA-seq library preparation and sequencing

Lin⁻CD25⁻CD44⁻ DN4 cells were sorted from the thymi of *LckCre⁺Zfp335⁺/⁺* and *LckCre⁺Zfp335^{fl/fl}* mice. The DN4 cell numbers in each group were as followed: WT1, $8 \times 10^5$ cells pulled from 4 mice; WT2, $9 \times 10^5$ cells from 5 mice; KO1, $6.7 \times 10^5$ cells from 13 mice; KO2, $7 \times 10^5$ cells from 15 mice. RNA isolation was performed using the RNeasy Mini Kit (Qiagen) according to the manufacturer's protocol. RNA quality and quantity were detected by the Qubit RNA broad range assay in the Qubit Fluorometer (Invitrogen). After quality control using RNase-free agarose gel and Agilent 2100 (Agilent Technologies, Palo Alto, CA, USA), RNA-seq libraries were prepared by using 200 ng total RNA with TruSeq RNA sample prep kit (Illumina). Oligo(dT)-enriched mRNAs were fragmented randomly with fragmentation buffer, followed by first- and second-strand cDNA synthesis. After a series of terminal repair, the double-stranded cDNA library was obtained through PCR enrichment and size selection. cDNA libraries were sequenced with the Illumina Hiseq 2000 sequencer (Illumina HiSeq 2000 v4 Single-Read 50 bp) after pooling according to its expected data volume and effective concentration.

Two biological replicates were performed in the RNA-seq analysis. Raw reads were then aligned to the mouse genome (GRCm38) using Tophat2 RNA-seq alignment software, and unique reads were retained to quantify gene expression counts from Tophat2 alignment files. Data were analyzed and preprocessed in the R environment. Differential expression analysis was performed using R package DESeq2 (adjusted p value < 0.05 and fold change >1.25). Heat maps and volcano plots were visualized using the R package.

## ChIP-seq library preparation and sequencing

Both Lin⁻CD25⁺CD44⁻ DN3 cells and Lin⁻CD25⁻CD44⁻ DN4 cells were sorted from the thymi of 100 *WT* mice by FACSAria flow cytometry (BD). Anti-Zfp335 antibody (Novus) and Millipore 17-10,085 ChIP kit were used in the ChIP assay. Immunoprecipitated DNA was used for Illumina ChIP-seq sample preparation. In brief, $5 \times 10^7$ cells were crosslinked to chromatin with 1% formaldehyde. Reaction was stopped with 0.125 M glycine. Cells were then resuspended in cold nuclear lysis buffer and sonicated to obtain DNA with ~300–500 bp size, followed by precipitation by incubation with immunoprecipitation-grade anti-Zfp335 antibody and Magnetic Protein A/G Beads overnight. The following day, beads were sequentially washed by low-salt, high-salt, LiCl, and TE buffers. Bound complexes were eluted in 150 µl of elution buffer at 62°C for 2 hr with shaking, followed by reversal of formaldehyde crosslinking at 95°C for 10 min. DNA was eventually purified with spin columns.

The concentration of immunoprecipitated DNA was detected by the Qubit DNA broad range assay in the Qubit Fluorometer (Invitrogen). 10 ng immunoprecipitated DNA was prepared for sequencing using the Illumina ChIP-seq sample preparation protocol. Blunt-end DNA fragments were ligated to Illumina adaptors, amplified, and sequenced using the SE150 model. Raw reads were filtered firstly to remove low-quality or adaptor sequences by SOAPnuke (version 1.5.6). Clean reads were mapped to the reference genome of GRCm39 with SOAPaligner/soap2 (version 2.21t) using default settings. The MACS2 software (Version 2.1.1) was used to process peak calling. MACS (Model-based Analysis of ChIP-Seq) is a commonly used computational method which is designed to identify peaks from ChIP-seq data. MACS assigns every candidate region an enrichment p value, and targeted genes are identified as final peaks passing the threshold p value < 1e−5 (*Feng et al., 2012*; *Feng et al., 2011*). The different enrichment peaks from different samples were plotted by MAnorm (version 1.1). Genomic graphs were generated and viewed with the IGV (Integrative Genomics Viewer).

## Statistical analysis

Data were presented as mean ± standard error of the mean. Statistical analyses were applied to biologically independent mice or technical replicates for each experiment which was independently repeated at least three times. Two-tailed Student's $t$-test was used for all statistical calculations using GraphPad Prism seven software. All bar graphs include means with error bars to show the distribution of the data. The level of significance is indicated as follows: *$p < 0.05$, **$p < 0.01$, ***$p < 0.001$, and ****$p < 0.0001$.

## Acknowledgements

We thank Drs. Xiaofei Wang and Guohua Zhang for Flow cytometry analysis and cell sorting. We also thank the Mouse Facility of Xi'an Jiaotong University.

## Additional information

### Funding

| Funder | Grant reference number | Author |
|---|---|---|
| National Natural Science Foundation of China | 3217080356 | Baojun Zhang |
| National Natural Science Foundation of China | 81771673 | Baojun Zhang |
| Major International Joint Research Programme | 81820108017 | Baojun Zhang |

The funders had no role in study design, data collection, and interpretation, or the decision to submit the work for publication.

### Author contributions

Xin Wang, Formal analysis, Investigation, Validation, Visualization, Writing – original draft, Writing – review and editing; Anjun Jiao, Investigation, Writing – original draft; Lina Sun, Investigation, Validation, Writing – original draft, Writing – review and editing; Wenhua Li, Investigation, Validation, Writing – original draft; Biao Yang, Yanhong Su, Renyi Ding, Haiyan Liu, Xiaofeng Yang, Chenming Sun, Investigation; Cangang Zhang, Formal analysis, Software; Baojun Zhang, Conceptualization, Funding acquisition, Project administration, Supervision, Visualization, Writing – original draft, Writing – review and editing

### Author ORCIDs

Xin Wang ⓘ http://orcid.org/0000-0002-1624-8241
Baojun Zhang ⓘ http://orcid.org/0000-0001-5972-1011

### Ethics

All mice were housed in specific-pathogen-free conditions by the Xi'an Jiaotong University Division of Laboratory Animal Research. All animal procedures were approved by the Institutional Animal Care and Use Committee of Xi'an Jiaotong University (2017-1012).

### Decision letter and Author response

Decision letter https://doi.org/10.7554/eLife.75508.sa1
Author response https://doi.org/10.7554/eLife.75508.sa2

## Additional files

### Supplementary files

- Supplementary file 1. Summary for down- and upregulated genes induced by Zfp335 deficiency.
- Supplementary file 2. ChIP-seq analysis of Zfp335-binding peaks in DN thymocytes.
- Supplementary file 3. Overlay analysis between downregulated genes from RNA-seq data and

Zfp335-binding genes from ChIP-seq data.
- Supplementary file 4. Primers used for quantitative PCR.
- Transparent reporting form

## Data availability

The sequencing data presented in this paper are available for download on GEO data repository with accession numbers GSE184532 and GSE184705. Source data files have been provided for relevant figures.

The following datasets were generated:

| Author(s) | Year | Dataset title | Dataset URL | Database and Identifier |
|---|---|---|---|---|
| Wang X, Jiao AJ, Sun LN, Wh LI, Yang B, Yh SU, Ding RY, Zhang CG, Liu HY, Yang XF, Sun CM, Zhang BJ | 2021 | Zinc finger protein Zfp335 controls thymocyte differentiation and survival through b-selection-dependent and -independent mechanisms | https://www.ncbi.nlm.nih.gov/geo/query/acc.cgi?acc=GSE184532 | NCBI Gene Expression Omnibus, GSE184532 |
| Wang X, Jiao AJ, Sun LN, Wh LI, Yang B, Yh SU, Ding RY, Zhang CG, Liu HY, Yang XF, Sun CM, Zhang BJ | 2021 | Zinc finger protein Zfp335 controls thymocyte differentiation and survival through b-selection-dependent and -independent mechanisms | https://www.ncbi.nlm.nih.gov/geo/query/acc.cgi?acc=GSE184705 | NCBI Gene Expression Omnibus, GSE184705 |

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
