## [Editor Report]

The authors have discovered that the transcription factor Zfp335 is an important regulator of early T cell development in the thymus. This paper will be of interest to scientists within the field of T cell development. The authors show that Bcl6 and Rorc are direct gene targets of Zfp335 and dysregulation of these are at least partly responsible for the impaired T cell development in Zfp335 mice.

---

## [Decision Letter]

**Decision letter after peer review:**

Thank you for submitting your article "Zinc finger protein Zfp335 controls early T cell development and survival through β-selection-dependent and -independent mechanisms" for consideration by *eLife*. Your article has been reviewed by 3 peer reviewers, and the evaluation has been overseen by a Reviewing Editor and Tadatsugu Taniguchi as the Senior Editor. The reviewers have opted to remain anonymous.

Essential revisions:

1. Better characterise the functional relationship between Zfp335 and β selection.

2. Control for the possibility that overexpression of Zfp335, Bcl6 or Rorc caused the effects in a Zfp335-independent manner by expressing the genes in a similar manner in WT cells.

3. Confirm stage-specific expression of Zfp335 with antibody staining.

4. Include information on cell counts.

5. Show the effects on pTα gene expression.

6. Explain how the downstream target genes were validated and triaged, including whether Bcl6 and RORc stood out from a bioinformatics perspective, or were selected based upon their known biology.

7. Justify the rationale for a 1:4 ratio of WT:KO

*Reviewer #1 (Recommendations for the authors):*

a) Overall the figures were clear and well laid out. However, it was sometimes quite complicated to decipher why there was a difference in the trends between % cells and cell counts (e.g. DN cells Figure 1 D versus E). I presume that this is because Zfp335 deficiency caused a drop in overall cell number but it might be helpful to show the overall cell counts data to help the reader interpret the data. The cell counts data should also be included in figure 2.

b) In the paper you identified a number of processes that were impacted by this transcription factor but did not look at the interaction between them. For instance, you demonstrated that intracellular TCRβ is reduced and apoptosis is increased by Zfp335 deficiency, and that forced expression of intracellular TCRβ partially elevated the effect on development but did not examine the effect on apoptosis. This would of be particularly interesting to examine given the recent work showing that pre-TCR increases the expression of Bcl6, which in turn protects DN4 cells from apoptosis (Development 2020 Oct 7;147(19):dev192203. doi: 10.1242/dev.192203).

c) The discussion would benefit from more discussion of the literature. I appreciate that there is only a limited amount of literature on the role of Zfp335, apart from the paper that you already mentioned, however you could discuss what is known about the role of this transcription factor in other developmental systems. Furthermore, it would be interesting to discuss how Zfp335 regulatory pathways compares to other regulatory pathways already known to control T cell development and how it might interplay.

d) In the discussion you state "While Zfp335-deficient DN4 cells exhibited no defects in the rearrangement of TCRβ chain genes and pTα gene expression". My apologies if I missed it but I could find the pTα data

*Reviewer #2 (Recommendations for the authors):*

Overall this manuscript starts with an interesting developmental phenotype however there are several problems with the approaches and experimental details that I indicated above. These include: (a) disregarding experimental issues like assessing DN3a versus DN3b, which would have located the initiation of the defect, (b) failing to explain approaches like the 1 to 4 ratio of WT to KO in the mixed cultures and chimeras, (c) specifically focusing on downregulated genes, (d) failing to explain how they narrow down to 97+22 genes from the original 566 downregulated and 899 upregulated genes when they compare with the genes associated with ChIP-seq peaks, (e) failing to explain how they identify the genes associated with each peak (f) failing to control how the overexpression of Rorc and Bcl6 affects the development of WT cells. In view of all these widespread issues, this reviewer has serious problems with the interpretations and mechanistic conclusions of the manuscript

*Reviewer #3 (Recommendations for the authors):*

This is an interesting study from Wang et al. The data presented is strong, thorough, well-controlled and largely support the conclusions made. However, I do have some suggestions that the authors should attempt to address.

1. While the qPCR analysis for Zfp335 expression validates the microarray data from ImmGen, this of course is still only RNA analysis. If would be good to validate the expression of Zfp335 protein, either by intracellular staining + FACS or by Western blotting of DN, DP and SP sorted cells. Commercial antibodies are available.

2. The authors showed that there is increased apoptosis of DN3 cells. Is this occurring pre-β-selection (DN3a) or post-β-select (DN3b)? Co-staining for additional markers, e.g. intracellular TCRbeta would address this. This is relevant given that authors suggest that the block is due to impaired i.c.TCRbeta expression.

3. Initially, the authors suggest that the block in Zfp335 knockout mice is due to impaired TCRbeta expression and that this can be rescued by the OT-1 transgene. However, they then suggest that Zfp335 functions by regulating Bcl6 and Rorc. How do the authors connect these two observations? Is the TCRb impairment a result of dysregulated Bcl6 and/or Rorc expression?

4. Bcl6 is normally expressed in DN3 thymocytes and regulates survival at that stage, while Rorc is normally expressed in DP thymocytes and regulates survival at that stage. The timing of Zfp335 expression aligns with Bcl6 but not with Rorc. How do the authors reconcile this discordance given that Zfp335 is downregulated in DP thymocytes?

---

## [Author Response]

Essential revisions:1. Better characterise the functional relationship between Zfp335 and β selection.

Please refer to the answers for Question b from Reviewer #1 and Question 2-3 from Reviewer #3.

2. Control for the possibility that overexpression of Zfp335, Bcl6 or Rorc caused the effects in a Zfp335-independent manner by expressing the genes in a similar manner in WT cells.

Please refer to the answer for Question f from Reviewer #2.

3. Confirm stage-specific expression of Zfp335 with antibody staining.

Please refer to the answer for Question 1 from Reviewer #3.

4. Include information on cell counts.

Please refer to the answer for Question a from Reviewer #1.

5. Show the effects on pTα gene expression.

Please refer to the answer for Question d from Reviewer #1.

6. Explain how the downstream target genes were validated and triaged, including whether Bcl6 and RORc stood out from a bioinformatics perspective, or were selected based upon their known biology.

We have performed overlay analysis between Zfp335 ChIP-seq (DN3/4 cells) and 119 profoundly downregulated genes (Zfp335 deficient DN4 cells vs WT DN4 cells). Ten out of 22 candidate genes were pursued to functional test. Meanwhile, we used pathway analysis to screen out 23 genes related to T cell development and/or survival. Based on known biology in thymocyte development, we decided to focus on Bcl6 and Rorc. Please see the description in the result part. Please also refer to the answer for Question c-d from Reviewer #2.

7. Justify the rationale for a 1:4 ratio of WT:KO

Please refer to the answer for Question b from Reviewer #2.

Reviewer #1 (Recommendations for the authors):a) Overall the figures were clear and well laid out. However, it was sometimes quite complicated to decipher why there was a difference in the trends between % cells and cell counts (e.g. DN cells Figure 1 D versus E). I presume that this is because Zfp335 deficiency caused a drop in overall cell number but it might be helpful to show the overall cell counts data to help the reader interpret the data. The cell counts data should also be included in figure 2.

We thank the reviewer for this advice. Indeed, the difference in the trends between % cells and cell counts in Figure 1D and 1E was a result of a dramatic reduction of overall cell number in the thymus caused by Zfp335 deficiency (Figure 1B). Following the suggestion, we have added the cell counts data for separate culture and separate transfer experiments, as well as ratio analysis for co-culture and co-transfer experiment in Figure2—figure supplement 1 of the revised version.

b) In the paper you identified a number of processes that were impacted by this transcription factor but did not look at the interaction between them. For instance, you demonstrated that intracellular TCRβ is reduced and apoptosis is increased by Zfp335 deficiency, and that forced expression of intracellular TCRβ partially elevated the effect on development but did not examine the effect on apoptosis. This would of be particularly interesting to examine given the recent work showing that pre-TCR increases the expression of Bcl6, which in turn protects DN4 cells from apoptosis (Development 2020 Oct 7;147(19):dev192203. doi: 10.1242/dev.192203).

Thank the reviewer for this interesting point. Following the suggestion, we examined the effect of forced expression of TCRβ on thymocyte apoptosis and the results revealed that DN3 and DN4 cells from OT1^Tg^KO mice exhibited similarly increased apoptosis with Zfp335 deletion (Figure5—figure supplement 4), suggesting Zfp335 affects thymocyte apoptosis in a TCR-independent manner.

In T cell development, β-selection promotes the transition from DN to DP cell, a process depending on the expression of pre-TCR signalling, which further provides signals for thymocyte proliferation and differentiation. In the Development 2020 paper (Solanki et al., 2020), it was reported that pre-TCR signaling induces Bcl6 expression which in turn inhibits DN4 apoptosis and promotes DN to DP transition. Whereas, increasing evidence have shown that TCR-independent signals also contribute to the survival and proliferation of thymocytes, such as IL7, Hedgehog and Wnt signalling (Staal et al., 2001; Outram et al., 2009; Rowbotham et al., 2009; Boudil et al., 2015).

In our study, Zfp335 is indispensable for thymocyte β-selection given that T cell-specific deficiency in Zfp335 leads to impaired intracellular TCRβ expression, blockade of thymocytes at DN stage as well as a substantial DN cell apoptosis. Though enhanced expression of TCRβ restores the developmental defect during DN3 to DN4 transition, it had little impact on the population size of DN3, DN4 and DP cells, suggesting the regulation of Zfp335 on DN cell apoptosis through mechanisms more than β-selection. Indeed, we provided the evidence that Zfp335 controlled DN cell survival through regulating Bcl6 and Rorc expression.

We have added these data in Figure5—figure supplement 4A-C and corresponding discussion in the revised manuscript.

c) The discussion would benefit from more discussion of the literature. I appreciate that there is only a limited amount of literature on the role of Zfp335, apart from the paper that you already mentioned, however you could discuss what is known about the role of this transcription factor in other developmental systems. Furthermore, it would be interesting to discuss how Zfp335 regulatory pathways compares to other regulatory pathways already known to control T cell development and how it might interplay.

Thanks for the good suggestions. We have added comprehensive discussion about the role of Zfp335 in other developmental systems and how it interplays with other regulatory pathways in T cell development. The new discussion is attached as below.

“Zfp335 was previously demonstrated to be crucial for early embryonic development as homozygous deletion of this gene results in neonatal death (Garapaty et al., 2009). Conditional knockout of Zfp335 in neural system led to severely reduced cortical size and impaired neurogenesis. Mechanistically, ZNF335 is required for neural progenitor cell self-renewal and proliferation, and neuronal differentiation (Yang et al., 2012) and neuronal morphogenesis (Zhao et al., 2015). Besides, deficiency of naïve T cells in mice carrying a hypomorph allele of Zfp335 (Zfp335bloto) uncovered its role in immune system (Han et al., 2014). So far, there is still very limited information about the functions of Zfp335 in other aspects of immune system. Here we found that Zfp335 is absolutely required for multiple steps of early T cell development. Of note, it will be worth investigating whether and how Zfp335 is involved in the regulation of mature T cell differentiation and functions under static and immunized conditions in future.”

“Several key factors have been shown to involve in the regulation of T cell developmental process from DN3 to DP stages. Notch signaling is required for early T cell commitment and β-selection (Radtke et al., 2010; Ciofani and Zuniga-Pflucker, 2005; Hosokawa and Rothenberg, 2018), which is subsequently weaken by Bcl6 repression for the differentiation from DN to DP stage development (Solanki et al., 2020). Consistently, our results found that Zfp335 could directly target Bcl6, and Zfp335 deficiency led to decreased expression of Bcl6 and upregulation of Notch target genes such as Dtx1 and Notch1 (data not shown), which collectively contributing to the developmental block from DN to DP stage. In addition, Tcf1 plays a vital role in T cell lineage commitment since Tcf1-/- DN3 thymocytes failed to be progressed to DN4 and subsequent DP stage through regulating Bcl11b and GATA3 expression (Garcia-Perez et al., 2020). Bcl11b and GATA3 also differentially regulate the differentiation and survival of thymocytes at DN3, DN4 and subsequent ISP stages via TCR or/and survival signals (Inoue et al., 2006; Pai et al., 2003). However, we did not detect reduced expression of Tcf1, Bcl11b or GATA3 in Zfp335-deficient DN4 cells, suggesting that Zfp335 regulates DN to DP thymocyte development independent on these molecules”.

d) In the discussion you state "While Zfp335-deficient DN4 cells exhibited no defects in the rearrangement of TCRβ chain genes and pTα gene expression". My apologies if I missed it but I could find the pTα data.

The pTα expression was detected by qPCR in the Figure5—figure supplement 1 as its gene name ptcra. We have marked it in the revised manuscript to make it clear.

Reviewer #2 (Recommendations for the authors):Overall this manuscript starts with an interesting developmental phenotype however there are several problems with the approaches and experimental details that I indicated above. These include:a) Disregarding experimental issues like assessing DN3a versus DN3b, which would have located the initiation of the defect,

We thank the reviewer for pointing it out. To address this question, we examined the effect of Zfp335 deficiency on cell development of DN3a and DN3b. The results showed that the frequencies of both DN3a and DN3b were comparable between WT and Zfp335 deficient groups. However, both cell populations showed enhanced cell apoptosis in Zfp335 deficient cells, indicating that Zfp335 regulates thymocyte apoptosis in TCR-independent manners. Those data have been added in Figure4—figure supplement 2 in the revised manuscript.

b) Failing to explain approaches like the 1 to 4 ratio of WT to KO in the mixed cultures and chimeras,

Thank the reviewer for pointing out this detailed question. In the initial experiment, we tried co-culture of WT and KO DN3 cells at 1:1 ratio, however KO cells were rapidly competed out and was not ideally to perform further analysis. In this regard, we switched the ratio to 1:4 in the mixed culture and chimeras, and significantly lower percentage of DP cells was still observed in KO group.

c) Specifically focusing on downregulated genes,

Thanks for the suggestions. Given that zinc finger protein Zfp335 functions as a transcription factor to positively regulate gene expression, we primarily focused on genes downregulated after Zfp335 deficiency to explore Zfp335 target genes. As pointed out by the reviewer, we also analyzed the upregulated genes (Author response image 1) .

**Author response image 1. sa2fig1:** 

Gene ontology (GO) analysis of upregulated genes revealed pathways related to lymphocyte apoptosis and differentiation were highly enriched (Author response image 1) , and pro-apoptotic genes Bax, caspase8 and Fas were increased in Zfp335 deficient cells (Author response image 1). In agreement with the observation of Bcl6 downregulation in Zfp335 deficient cells, Dtx1, inhibited by Bcl6, was increased in Zfp335 deficient cells (Author response image 1). Since we aim to screen direct targets by Zfp335, we did not include these data in our manuscript.

d) Failing to explain how they narrow down to 97+22 genes from the original 566 downregulated and 899 upregulated genes when they compare with the genes associated with ChIP-seq peaks,

We apologize for the confusion about the gene number. In the RNA-seq data, 566 genes were downregulated using fold change>1.25 as a cutoff. We expected more genes were included for pathway enrichment analysis. To narrow down the Zfp335 targeting candidates, 119 (97+22) profoundly downregulated genes were further selected out by a cutoff of 2-fold change, and 22 genes were overlapped with Zfp335-targeting genes from ChIP-seq data. We have clarified it in the manuscript and figure legend in the revised manuscript.

e) Failing to explain how they identify the genes associated with each peak

In the ChIP-seq data analysis, the MACS2 software (Version 2.1.1) was used to process peak calling. MACS (Model-based Analysis of ChIP-seq) is a commonly used computational method which was designed to identify peaks from ChIP-seq data. MACS assigns every candidate region an enrichment p-value, and targeted genes were identified as final peaks passing the threshold p-value<1e-5 (Feng et al., 2012, Nat Protoc, Feng et al., 2011, Curr Protoc Bioinformatics). We have clarified it in the Method section and cited related literatures in the revised manuscript.

f) Failing to control how the overexpression of Rorc and Bcl6 affects the development of WT cells. In view of all these widespread issues, this reviewer has serious problems with the interpretations and mechanistic conclusions of the manuscript

As suggested, we have performed overexpression of Bcl6 and Rorc in WT DN3 cells and the results showed that overexpression of both Bcl6 and Rorc could increase DP thymocyte population, and overexpression of Rorc particularly increased CD4^+^ SP cells (Author response image 2) , which is in line with previous finding of essential role of Rorc in T lymphocyte maturation (He et al., 2000, J Immunol).

Reviewer #3 (Recommendations for the authors):This is an interesting study from Wang et al. The data presented is strong, thorough, well-controlled and largely support the conclusions made. However, I do have some suggestions that the authors should attempt to address.1. While the qPCR analysis for Zfp335 expression validates the microarray data from ImmGen, this of course is still only RNA analysis. If would be good to validate the expression of Zfp335 protein, either by intracellular staining + FACS or by Western blotting of DN, DP and SP sorted cells. Commercial antibodies are available.

We thank reviewer for the good suggestion. We have measured the Zfp335 protein expression among DN (DN3 and DN4), DP and SP by FACS. The results showed that consistent with mRNA data, Zfp335 has the highest expression in DN3 thymocytes. This data has been added in Figure1—figure supplement 1.

2. The authors showed that there is increased apoptosis of DN3 cells. Is this occurring pre-β-selection (DN3a) or post-β-select (DN3b)? Co-staining for additional markers, e.g. intracellular TCRbeta would address this. This is relevant given that authors suggest that the block is due to impaired i.c.TCRbeta expression.

Following the suggestions, we have performed experiments investigating how Zfp335 deficiency affects DN3a and DN3b cell population and their apoptosis. The results showed that the frequencies of both DN3a and DN3b were comparable between WT and Zfp335 deficient groups. However, both cell populations showed enhanced cell apoptosis in Zfp335 deficient cells, indicating that Zfp335 regulates thymocyte apoptosis in TCR-independent manners. Those data have been added in Figure4—figure supplement 2 in the revised manuscript.

3. Initially, the authors suggest that the block in Zfp335 knockout mice is due to impaired TCRbeta expression and that this can be rescued by the OT-1 transgene. However, they then suggest that Zfp335 functions by regulating Bcl6 and Rorc. How do the authors connect these two observations? Is the TCRb impairment a result of dysregulated Bcl6 and/or Rorc expression?

Thanks for the important question. We have performed overexpression of Bcl6 and Rorc in Zfp335 deficient DN3 thymocytes and the results showed that neither Bcl6 or Rorc overexpression could restore the decreased intracellular TCRβ expression in OP9-DL1 culture, suggesting Zfp335 regulates TCRβ expression independent on Bcl6 and Rorc. We have added these data in Figure 7—figure supplement 3 in the revised manuscript.

As our response to Reviewer 1, Zfp335 is indispensable for thymocyte β-selection given that T cell-specific deficiency in Zfp335 leads to impaired intracellular TCRβ expression and blockade of thymocytes at DN stage. Enhanced expression of TCRβ restored the developmental defect during DN3 to DN4 transition, but had little impact on the population size of DN3, DN4 and DP cells. These data clearly support the regulation of Zfp335 on DN cell apoptosis through mechanisms more than β-selection. Zfp335 improves DN thymocyte survival by directly regulating Bcl6 and Rorc expression. We also added discussion to elucidate the relations on Zfp335, TCRβ, Bcl6 and Rorc.

4. Bcl6 is normally expressed in DN3 thymocytes and regulates survival at that stage, while Rorc is normally expressed in DP thymocytes and regulates survival at that stage. The timing of Zfp335 expression aligns with Bcl6 but not with Rorc. How do the authors reconcile this discordance given that Zfp335 is downregulated in DP thymocytes?

Indeed, Zfp335 has the highest expression in DN3 thymocytes both in mRNA and protein level, suggesting its potent role in DN cell development. Indeed, Rorc expression was significantly downregulated in Zfp335 KO DN4 cells, and Rorc overexpression promoted the DN to DP transition.

However, downregulated Zfp335 in DP thymocytes is still important for DP cell survival as overexpression of Zfp335 rescued DP thymocytes from enhanced apoptosis (Figure 7F). In addition, Zfp335 regulates DP thymocyte development through Rorc as Rorc overexpression could rescue DP cell apoptosis and DP cell population in Zfp335 deficient cells. Therefore, compared to DN cells, DP cells express relative low level of Zfp335, which still indispensably participate the regulation of Rorc expression.